# Causal Discovery with Deductive Reasoning: One Less Problem

**Jonghwan Kim**[1]      **Inwoo Hwang**[2]      **Sanghack Lee**[* 1,2]

[1]Graduate School of Data Science, Seoul National University, South Korea
[2]Artificial Intelligence Institute, Seoul National University, South Korea
[*]Correspondence to: `sanghack@snu.ac.kr`

## Abstract

Constraint-based causal discovery algorithms aim to extract causal relationships between variables of interest by using conditional independence tests (CITs). However, CITs with large conditioning sets often lead to unreliable results due to their low statistical power, propagating errors throughout the course of causal discovery. As the reliability of CITs is crucial for their practical applicability, recent approaches rely on either tricky heuristics or complicated routines with high computational costs to tackle inconsistent test results. Against this background, we propose a principled, simple, yet effective method, coined DEDUCE-DEP, which corrects unreliable conditional independence statements by replacing them with deductively reasoned results from lower-order CITs. An appealing property of DEDUCE-DEP is that it can be seamlessly plugged into existing constraint-based methods and serves as a modular subroutine. In particular, we showcase the integration of DEDUCE-DEP into representative algorithms such as HITON-PC and PC, illustrating its practicality. Empirical evaluation demonstrates that our method properly corrects unreliable CITs, leading to improved performance in causal structure learning.

## 1 INTRODUCTION

One of the fundamental tasks in the realm of scientific inquiry is to extract cause-and-effect relationships among diverse variables of interest. Typically, causal relationships are often elucidated through randomized experiments. However, there are instances where extracting causal relationships through experiments is not feasible due to cost or ethical considerations. In such cases, causal discovery from observational data is crucial. [Pearl, 2009, Spirtes et al., 2000].

Causal discovery is the process of learning the underlying causal relationships in the form of a directed acyclic graph (DAG) typically from observational data. So far, several methodologies related to this task have been proposed.

Among the well-known causal discovery methodologies is the constraint-based method, represented by the PC algorithm [Spirtes et al., 2000]. It involves sequentially performing conditional independence tests (CITs) in a principled manner and synthesizing the results to induce the equivalence class of underlying causal structure [Meek, 1995, Dor and Tarsi, 1992]. Therefore, the reliability of CITs is crucial in this methodology. Similarly, the algorithmic correctness of many constraint-based methods is contingent on the critical assumption that all CITs performed on the data are correct. However, an oracle CIT rarely, if ever, exists in the real world.

In practice, CITs are prone to errors, particularly when dealing with large conditioning sets. As the size of the conditioning set increases, the number of data instances required grows exponentially, and thus, the statistical power of the corresponding CIT decreases. Such high-order CITs frequently lead to unreliable results, propagating errors throughout the process of structure learning. This contributes to the instability and performance degradation of the algorithm [Spirtes et al., 2000, Aliferis et al., 2010b, Armen and Tsamardinos, 2014].

Recent approaches to addressing the reliability concerns of CITs utilize rules derived from graphoid axioms [Geiger, 1990, Pearl and Paz, 1987] to build a causal structure that is as consistent as possible given the inconsistent CIT results [Bromberg and Margaritis, 2009, Ma et al., 2023]. The intuition is that graphoid axioms can be used to constrain conditional independence (CI) statements by other CI statements. However, they lack a principled way of determining the preference of CI statements and their practical applicability is often limited due to the high computational cost of searching possible combinations of rules from graphoid axioms.

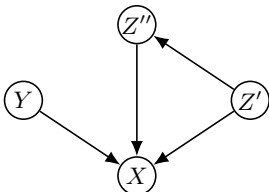

Figure 1: A subgraph of a true causal graph.

Against this background, this paper presents a straightforward, practical approach to causal discovery based on deductive reasoning over CI statements and graphoid axioms. Our method, coined DEDUCE-DEP, offers a simple and sound condition for correcting CITs, in contrast to previous approaches relying on complex routines. Specifically, our method replaces unreliable high-order CIT with outcomes derived from deductive reasoning with lower-order CITs and rules from graphoid axioms. Notably, our method serves as a modular subroutine that can be seamlessly integrated into various constraint-based methods, highlighting its practicality.

**Motivating example.** Consider the following scenario where a constraint-based method tries to discover the structure illustrated in Fig. 1. Due to the behavior of the algorithm, it tries to examine the adjacency between $X$ and $Y$ where the following CI statements are accurately obtained

$$(X \not\perp\!\!\!\perp Y \mid Z') \text{ and } (X \not\perp\!\!\!\perp Y \mid Z'').$$

Unfortunately, the relationship between $X$ and $Y$ is relatively weak, and we wrongly obtained

$$(X \perp\!\!\!\perp Y \mid Z', Z''),$$

which usually grants the removal of the true edge between $X$ and $Y$. We may have a doubt about the CI result, worrying about its power being low. We then examine the following CI between $Y$ and $Z''$ given $Z'$, where the CIT correctly yields $(Y \perp\!\!\!\perp Z'' \mid Z')$. In such case, we can indeed induce $(X \not\perp\!\!\!\perp Y \mid Z', Z'')$ from the previous CIT results and $(Y \perp\!\!\!\perp Z'' \mid Z')$ via applying rules derived from graphoid axioms, which will falsify the suspicious result, $(X \perp\!\!\!\perp Y \mid Z', Z'')$. Here, we prefer deductively reasoned $(X \not\perp\!\!\!\perp Y \mid Z', Z'')$ to the CIT result $(X \perp\!\!\!\perp Y \mid Z', Z'')$ given that tests with a smaller number of conditioning sets are *likely* more reliable.

**Contributions.** We propose a practical correction method for unreliable CITs using deductive reasoning. Our contributions are as follows. 1) We devised conditions and rules for deducing high-order CI statement with low-order CI statements. We developed DEDUCE-DEP, a sound algorithm that implements these rules in a recursive manner. The algorithm corrects unreliable independence statements with the (in)dependence statements from deductive reasoning. 2) Our correction algorithm can be *effortlessly* adapted to

any constraint-based causal discovery algorithm as a subroutine. In particular, we presented HITON-PC [Aliferis et al., 2010a] and PC [Spirtes et al., 2000] equipped with DEDUCE-DEP. 3) We examined our method with comprehensive experiments using both synthetic and semi-synthetic datasets. Empirical evaluation reveals that our method properly rectifies the consequences of unreliable CITs, improving the performance of causal structure learning.

## 2 PRELIMINARIES

We denote a variable by a capital letter $X$ and its realized value by the lowercase $x$ in the domain $D_X$. A set of variables will be expressed by a bold letter $\mathbf{X}$. We denote by $\mathbf{X} \perp\!\!\!\perp \mathbf{Y} \mid \mathbf{Z}$ the CI statement that the random variables $\mathbf{X}$ and $\mathbf{Y}$ are conditionally independent given a set of variables $\mathbf{Z}$. Similarly, we use $\mathbf{X} \not\perp\!\!\!\perp \mathbf{Y} \mid \mathbf{Z}$ to express conditional dependence. We denote a CI query by $(\mathbf{X}; \mathbf{Y} \mid \mathbf{Z})$, which can be either true (independent) or false (dependent).

A Bayesian network (BN) [Neapolitan, 1990, Pearl, 2000] is a tuple $(\mathcal{G}, P)$ where the joint distribution $P$ over a set of variables $\mathbf{V}$ is represented by the directed acyclic graph (DAG) $\mathcal{G}$. We assume that the underlying causal mechanisms can be encoded using a directed graphical model. A DAG $\mathcal{G}$ is defined as a tuple $(\mathbf{V}, \mathbf{E})$ where $\mathbf{V}$ and $\mathbf{E}$ are the set of all vertices and edges between pairs of vertices, respectively. We employ graphical kinship terminology $Pa(\cdot)$ and $Ch(\cdot)$ to represent parents and children in $\mathcal{G}$, respectively. Further, $Ne(\cdot)$ represents neighbors, i.e., nodes adjacent to the argument in an (un)directed graph.

### 2.1 CONSTRAINT-BASED CAUSAL DISCOVERY

Constraint-based causal discovery methods utilize multiple CITs to uncover causal relations from data. These methods are flexible since they do not impose any functional assumptions on the underlying causal relations. Prominent algorithms in this category include the PC (Peter-Clark) algorithm [Spirtes et al., 2000], which is sound and complete to discover the completed partially DAG (CPDAG). FCI [Spirtes et al., 2000] also belongs to this group but relaxes causal sufficiency, i.e., no unmeasured confounder.

Besides, there are local structure learning algorithms that learn the Markov blanket or parent-children set of a target variable from data [Yu et al., 2020]. In practice, such algorithms have shown remarkable scalability with thousands of variables [Aliferis et al., 2010a]. IAMB [Tsamardinos and Aliferis, 2003] and PCMB [Pena et al., 2007] are some of the renowned examples of Markov blanket learning algorithm, whereas MMPC [Aliferis et al., 2010a] and HITON-PC[1] [Aliferis et al., 2010a] are for parents-children learning.

---

[1] HITON comes from $\chi\iota\tau\acute{\omega}\nu$ (chiton), a form of fabric (blanket). Here, PC stands for Parents and Children, not Peter-Clark.

Constraint-based methods vary in multiple aspects, e.g., their underlying assumptions, types of output, strategies, and inductive biases employed to construct causal structures. For instance, the PC algorithm initiates learning from a complete undirected graph, while the MMPC and HITON-PC algorithms start from an empty graph. Despite these differences, these algorithms commonly encounter issues with unreliable CITs and the subsequent negative impacts on the learning process, posing a significant challenge in the task of causal discovery [Claassen and Heskes, 2012].

## 2.2 RELIABILITY CRITERION

So far, several constraint-based methods have introduced a few methods to properly tackle the reliability issue of CITs. The majority of these methods are based on heuristics. Some prominent examples are *heuristic power rule* and *the degree of freedom adjustment heuristic* [Tsamardinos and Borboudakis, 2010], which are adopted by PC and MMHC algorithm [Tsamardinos et al., 2006].

The heuristic power rule prescribes that a statistical test can be considered reliable and should be conducted only if a *sufficient* number of data instances are available per parameter, i.e., the number of cells in the contingency tables required for the CIT. If not, the statistical test does not proceed. In this way, the heuristic power rule implicitly considers the decision of the latest CIT performed as more reliable than that of the current test. As a result, when tests with higher-order conditioning sets are omitted, algorithms like HITON-PC assume dependence between the variables, delegating their decisions to the latest CIT performed. The threshold for the heuristic power rule varies across different causal discovery algorithms (e.g., 5 in Tsamardinos et al. [2006], or 10 in Spirtes et al. [2000]). However, an issue arises from the uncertainty surrounding how many instances are truly sufficient for accurate statistical inference.

The degree of freedom adjustment heuristic calibrates the degree of freedom (DoF) with respect to *zero* counts that appear in the contingency table when performing CITs. These zero counts may come from either random events or structural constraints of data generating process where the latter is called *structural zero*. Since structural zeros are not free to vary, the DoF should be recalculated in a way that subtracts one for each structural zero. A reduced DoF is expected to increase the power of the CIT. However, the problem is that we may not know which zero count is structural zero.

In addition to the previously discussed heuristics, there is another heuristic that involves limiting the size of the conditioning set when performing CITs [Tsamardinos et al., 2006, Aliferis et al., 2010a]. This prevents certain CITs from being performed if their conditioning set size exceeds a user-specified threshold. The behavior of this heuristic is akin to that of the heuristic power rule, since it skips some CITs

to avoid potential false negatives. Several implementations of constraint-based algorithms (e.g., causal-learn Python package [Zheng et al., 2024]) have implicitly adopted this heuristic to prematurely terminate the structure learning process. Despite its frequent adoption, a comprehensive understanding of the outcomes generated by these implementations has been lacking so far. Only recently has the understanding of this heuristic become a subject of ongoing research [Kocaoglu, 2024].

## 2.3 RULES FROM GRAPHOID AXIOMS

We introduce rules derived from *graphoid axioms*, which govern the relationships between CI statements [Pearl and Paz, 1987, Lauritzen, 1996, Paz et al., 1997]. The rules play a crucial role as integrity constraints, aiding in the resolution of conflicts arising from inconsistent CI statements [Bromberg and Margaritis, 2009]. Among several rules (see Appendix A), we will make use of the following rules:

1. Symmetry:
$(\mathbf{X} \perp\!\!\!\perp \mathbf{Y} | \mathbf{Z}) \Leftrightarrow (\mathbf{Y} \perp\!\!\!\perp \mathbf{X} | \mathbf{Z})$

2. Contraction:
$(\mathbf{X} \perp\!\!\!\perp \mathbf{Y} | \mathbf{Z}) \wedge (\mathbf{X} \perp\!\!\!\perp \mathbf{W} | \mathbf{Z}, \mathbf{Y}) \Rightarrow (\mathbf{X} \perp\!\!\!\perp \mathbf{Y}, \mathbf{W} | \mathbf{Z})$

3. Decomposition:
$(\mathbf{X} \perp\!\!\!\perp \mathbf{Y}, \mathbf{W} | \mathbf{Z}) \Rightarrow (\mathbf{X} \perp\!\!\!\perp \mathbf{Y} | \mathbf{Z}) \wedge (\mathbf{X} \perp\!\!\!\perp \mathbf{W} | \mathbf{Z})$

4. Weak transitivity:
$(\mathbf{X} \perp\!\!\!\perp \mathbf{Y} | \mathbf{Z}) \wedge (\mathbf{X} \perp\!\!\!\perp \mathbf{Y} | \mathbf{Z}, W) \Rightarrow (\mathbf{X} \perp\!\!\!\perp W | \mathbf{Z}) \vee (W \perp\!\!\!\perp \mathbf{Y} | \mathbf{Z})$

Unlike others, weak transitivity holds only under the assumption of a faithful Bayesian network $(\mathcal{G}, P)$, in which the graph $\mathcal{G}$ encodes all the CI information from the probability distribution $P$. Also note that $W$ in weak transitivity is a single variable. In the sequel, these rules will be useful for our approach to causal discovery based on deductive reasoning over CI statements.

## 3 METHOD

In this work, we make two assumptions widely adopted in the literature: causal Markov assumption and faithfulness, which play crucial roles in connecting probability distribution with a graph structure. Causal Markov assumption states that if d-separation holds in a causal graph, then the corresponding CI holds in the associated probability distribution. Faithfulness assumption ensures that all the observed CI in the probability distribution can be read off the corresponding causal graph. For the demonstration, we utilize constraint-based methods which assume causal sufficiency.

As discussed earlier, low-power CITs produce unreliable CI statements, leading to errors and instability in the structure learning. Although several heuristics have been proposed to tackle this issue, they either lack a theoretical basis or fall

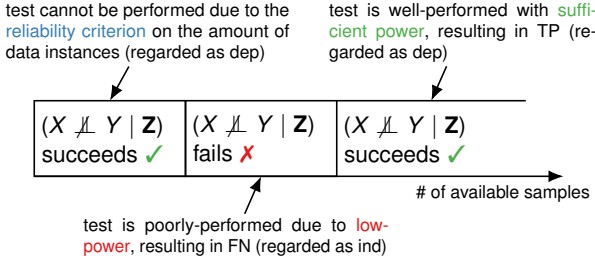

test cannot be performed due to the reliability criterion on the amount of data instances (regarded as dep)

test is well-performed with sufficient power, resulting in TP (regarded as dep)

| $(X \not\perp\!\!\!\perp Y \mid \mathbf{Z})$ succeeds ✓ | $(X \not\perp\!\!\!\perp Y \mid \mathbf{Z})$ fails ✗ | $(X \not\perp\!\!\!\perp Y \mid \mathbf{Z})$ succeeds ✓ |
|---|---|---|

# of available samples

test is poorly-performed due to low-power, resulting in FN (regarded as ind)

Figure 2: Behavior of HITON-PC and PC with heuristic power rule (adapted from Tsamardinos et al. [2006]).

---

**Algorithm 1** DEDUCE-DEP

1: **Input**: $\{X\}, \{Y\}, \mathbf{Z}$ disjoint subsets of $\mathbf{V}$, reliability threshold $K$ (default 1)
2: **Output**: Whether $(X \perp\!\!\!\perp Y \mid \mathbf{Z})$ is deducible or not.
3: **if** $|\mathbf{Z}| \leq K$ **return** FALSE
4: **for** $Z \in \mathbf{Z}$
5: $\quad \mathbf{Z}' \leftarrow \mathbf{Z} \setminus \{Z\}$
6: $\quad$ **for** $(\mathbf{A}, \mathbf{B}, \mathbf{C})$ in $\{(X, Y, \mathbf{Z}'), (X, Z, \mathbf{Z}'), (Y, Z, \mathbf{Z}')\}$
7: $\quad\quad$ **if** $(\mathbf{A} \perp\!\!\!\perp \mathbf{B} \mid \mathbf{C})$ **and not** DEDUCE-DEP$(\mathbf{A}, \mathbf{B}, \mathbf{C})$
8: $\quad\quad\quad$ mark $(\mathbf{A}; \mathbf{B} \mid \mathbf{C})$ as $\perp\!\!\!\perp$
9: $\quad\quad$ **else** mark $(\mathbf{A}; \mathbf{B} \mid \mathbf{C})$ as $\not\perp\!\!\!\perp$
10: $\quad$ **if** $(X \not\perp\!\!\!\perp Y \mid \mathbf{Z}') \oplus \big((X \not\perp\!\!\!\perp Z \mid \mathbf{Z}') \wedge (Y \not\perp\!\!\!\perp Z \mid \mathbf{Z}')\big)$
11: $\quad\quad$ **return** TRUE
12: **return** FALSE

---

### 3.1 RULES FOR DEDUCTIVE REASONING

In the following proposition, we present the rules for our reasoning method. Omitted proof is provided in Appendix B.

**Proposition 1.** *Under the faithful Bayesian network* $(\mathcal{G}, P)$, *let* $\mathbf{X}$, $\mathbf{Y}$, *and* $\mathbf{Z}$ *be disjoint subsets of* $\mathbf{V}$ *where* $\mathbf{Z}$ *is partitioned into* $\mathbf{Z}'$ *and* $\mathbf{Z}''$ *such that* $\mathbf{Z} = \mathbf{Z}' \sqcup \mathbf{Z}'', |\mathbf{Z}''| = 1$. *Then,* $(\mathbf{X} \not\perp\!\!\!\perp \mathbf{Y} \mid \mathbf{Z})$ *if one of the following holds:*

1. $(\mathbf{X} \not\perp\!\!\!\perp \mathbf{Y} \mid \mathbf{Z}') \wedge (\mathbf{X} \perp\!\!\!\perp \mathbf{Z}'' \mid \mathbf{Z}')$
2. $(\mathbf{X} \perp\!\!\!\perp \mathbf{Y} \mid \mathbf{Z}') \wedge (\mathbf{X} \not\perp\!\!\!\perp \mathbf{Z}'' \mid \mathbf{Z}') \wedge (\mathbf{Y} \not\perp\!\!\!\perp \mathbf{Z}'' \mid \mathbf{Z}')$

These rules provide a theoretical background for our deductive reasoning-based approach. They stipulate several conditions where we can deduce high-order dependence statements from strictly low-order CI statements. Note that the first condition in Prop. 1 can be modified to

$$(\mathbf{X} \not\perp\!\!\!\perp \mathbf{Y} \mid \mathbf{Z}') \wedge (\mathbf{Y} \perp\!\!\!\perp \mathbf{Z}'' \mid \mathbf{Z}')$$

by the symmetry, which we will make use of it in the sequel. We now proceed to incorporate these rules into an algorithm.

### 3.2 INCORPORATING DEDUCTIVE REASONING RULES INTO ALGORITHM

We propose DEDUCE-DEP (Alg. 1), a sound algorithm for deducing dependence statements from strictly low-order CI statements. In particular, it can be viewed as a special case of Prop. 1 where $\mathbf{X} = \{X\}$ and $\mathbf{Y} = \{Y\}$.

As DEDUCE-DEP is designed to examine CI statements, arguments of the algorithm are $X$, $Y$, and $\mathbf{Z}$ which constitute the statement $(X \perp\!\!\!\perp Y \mid \mathbf{Z})$. We let the size of the minimal conditioning set as a hyperparameter $K$, which specifies the base case for recursive calls. We assume a marker (a cache keeping CIT results) is globally defined. As an output, DEDUCE-DEP returns whether a dependence statement $(X \not\perp\!\!\!\perp Y \mid \mathbf{Z})$ is attainable from strictly low-order CITs.

---

short of properly correcting unreliable CITs. For instance, the heuristic power rule considers only the amount of data available for the test to determine its reliability. However, as illustrated in Fig. 2, it leaves quite a few CITs with CI statements untouched, even though they are suspicious of being unreliable. This phenomenon is problematic because even with the number of data instances above the user-specified threshold, we can expect that CITs falsely declare independence due to their lack of statistical power. In this respect, our approach should aim possibly all the CI statements during the learning process. Putting pre-existing heuristics aside, *how can we properly tell whether CI statements from CITs are true?*

Inspiration for our method comes from the fact that the outcome of CIT can be constrained by the outcomes of other CITs through graphoid axioms [Pearl and Paz, 1987]. This insight opens the door for CI statement correction through deductive reasoning with the rules derived from graphoid axioms. It provides a sound way to reason about the CI query of interest from multiple other CI statements that we judge to be reliable. In this regard, we sketch our approach as follows. For every CI statement obtained, we conduct deductive reasoning to figure out whether a dependence statement can be logically induced from other low-order CITs. If it can be, we adopt the result from deductive reasoning and, if otherwise, the one from a statistical test.

In our approach, we restricted the ingredients of deductive reasoning to *lower-order* CIT, whose conditioning set is a proper subset of that of the CI query. Clearly, they are expected to be more reliable than CI statements of our interest since the amount of data needed for sound statistical inference is smaller. Therefore, we opt to the result from deductive reasoning whenever dependence is logically attainable from these lower-order tests. If needed, we add new lower-order CITs for our reasoning. As long as the CIT is lower-order, then the corresponding test result is eligible to be the ingredients of our deductive reasoning and should be performed. Allowing conducting more CITs, not just harnessing already executed, broadens the scope to which our deductive reasoning can be applied.

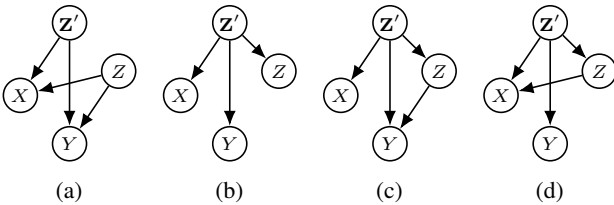

Figure 3: Examples for Prop. 2.

DEDUCE-DEP starts with checking the size of conditioning set $\mathbf{Z}$ (line 3 in Alg. 1). If it is smaller than the reliability threshold $K$, then DEDUCE-DEP returns FALSE, affirming the corresponding CIT result as reliable. If not, DEDUCE-DEP proceeds to partition $\mathbf{Z}$ into a singleton $\{Z\}$ and the complementary $\mathbf{Z}'$ (lines 4–5). Based on Prop. 1, it performs three CITs, $(X; Y \mid \mathbf{Z}'), (X; Z \mid \mathbf{Z}'), (Y; Z \mid \mathbf{Z}')$, to infer whether $(X \not\perp\!\!\!\perp Y \mid \mathbf{Z}', Z)$ is deducible or not (line 6), where the results are marked in marker. If the marked result is independence, then it recursively calls DEDUCE-DEP (lines 7–8) and mark the results either from CITs or deductive reasoning (lines 8–9). Here, marks can be updated from independence to dependence. If Prop. 1 can be applied based on the marks of three CI statements, it returns TRUE (lines 10–11). It repeats this procedure for all $Z \in \mathbf{Z}$ until it returns TRUE; otherwise, it returns FALSE (line 12).

In theory, our algorithm examines three CI statements to determine whether a dependence statement of our interest is deducible. However, in some cases, it can do so with just two CI statements. If we obtain $(X \not\perp\!\!\!\perp Y \mid \mathbf{Z}')$ and subsequently obtain $(X \perp\!\!\!\perp Z \mid \mathbf{Z}')$, then with these two CI statements, we can satisfy the conditions to deduce the dependence statement in Line 10. Similarly, if we obtain $(X \not\perp\!\!\!\perp Y \mid \mathbf{Z}')$ and then obtain $(Y \perp\!\!\!\perp Z \mid \mathbf{Z}')$, the same applies. This early stopping mechanism could be quite efficient in practical implementation as it reduces unnecessary recursive calls. We now proceed to show the correctness of DEDUCE-DEP.

**Corollary 1.** *Under the faithful Bayesian network* $(\mathcal{G}, P)$, *let* $\{X\} \sqcup \{Y\} \sqcup \mathbf{Z} \subseteq \mathbf{V}$, $Z \in \mathbf{Z}$, *and* $\mathbf{Z}' = \mathbf{Z} \setminus \{Z\}$. *If*

$$(X \not\perp\!\!\!\perp Y \mid \mathbf{Z}') \oplus \big( (X \not\perp\!\!\!\perp Z \mid \mathbf{Z}') \wedge (Y \not\perp\!\!\!\perp Z \mid \mathbf{Z}') \big),$$

*then* $(X \not\perp\!\!\!\perp Y \mid \mathbf{Z})$ *holds.*

Cor. 1 provides a sound condition for *deducing* the dependence given the three statements and claims nothing for the cases where the condition is false. We show the possibility of $(X \perp\!\!\!\perp Y \mid \mathbf{Z})$ for the cases that are not true in Cor. 1.

**Proposition 2.** *The condition in Cor. 1 is complete for deducing* $(X \not\perp\!\!\!\perp Y \mid \mathbf{Z})$ *with respect to the three conditional CI statements.*

*Proof.* We prove by constructing a counterexample for each tuple $((X; Y \mid \mathbf{Z}'), (X; Z \mid \mathbf{Z}'), (Y; Z \mid \mathbf{Z}'))$ where the

---

**Algorithm 2** HITON-PC-Nonsym with DEDUCE-DEP

1: **Input**: a set of variables $\mathbf{V}$, the target variable $T$, dependency measure, CI tester
2: **Output**: a tentative parents-children set TPC for $T$
3: Initialize TPC with an empty set.
4: Let OPEN be $\{X \in \mathbf{V} \mid X \not\perp\!\!\!\perp T\}$
5: Sort OPEN in descending order w.r.t correlation with $T$
6: **while** OPEN is not empty
7:     Let $X$ be an element popped from OPEN.
8:     Update TPC with $X$.
9:     **repeat**
10:       **for** $Y \in$ TPC
11:         **if** $(Y \perp\!\!\!\perp T \mid \mathbf{S})$ for some $\mathbf{S} \subseteq$ TPC $\setminus \{Y\}$
12:           **if** not DEDUCE-DEP($\{Y\}, \{T\}, \mathbf{S}$)
13:             Remove $Y$ from TPC.
14:     **until** No change in TPC
15: **return** TPC

---

condition is *not* satisfied: $(\not\perp\!\!\!\perp, \not\perp\!\!\!\perp, \not\perp\!\!\!\perp)$ in Fig. 3a, $(\perp\!\!\!\perp, \perp\!\!\!\perp, \perp\!\!\!\perp)$ in Fig. 3b, $(\perp\!\!\!\perp, \perp\!\!\!\perp, \not\perp\!\!\!\perp)$ in Fig. 3c, and $(\perp\!\!\!\perp, \not\perp\!\!\!\perp, \perp\!\!\!\perp)$ in Fig. 3d where $\mathbf{Z}' \to W$ implies $Z' \to W$ for every $Z' \in \mathbf{Z}'$. For all cases, we have $(X \perp\!\!\!\perp Y \mid \mathbf{Z})$. $\qquad\square$

# 4 APPLICATION OF DEDUCTIVE REASONING ON CAUSAL DISCOVERY

An appealing property of DEDUCE-DEP is that it can be seamlessly integrated into any constraint-based methods and serves as a modular subroutine. This highlights the practicality of DEDUCE-DEP since it aims to correct statistical errors from unreliable CITs, which could further prevent error propagation throughout the learning process. To elucidate how our method works inside the constraint-based methods, we take HITON-PC (Sec. 4.1) and PC algorithms (Sec. 4.2), representative local and global structure learning algorithms, respectively, as illustrative examples.

## 4.1 HITON-PC WITH DEDUCE-DEP

We augment a local structure learning algorithm, HITON-PC [Aliferis et al., 2010a], with DEDUCE-DEP as a subroutine. HITON-PC is an algorithmic instance from the Generalized Local Learning Parents and Children framework [Aliferis et al., 2010a], which outputs parents and children of a target variable. It consists of two subroutines, namely, HITON-PC-Nonsym and symmetry correction. The former outputs the superset of the parents-children set of a target variable, which is then pruned by the latter. Although pruning ensures the outcome of HITON-PC-NonSym is correct, empirical evaluations have shown that applying symmetry correction adds little to performance improvement, sometimes leading to performance degradation. This is due to the frequent occurrence of low-power CITs and their subsequent influences on the algorithm [Aliferis et al., 2010b].

---

**Algorithm 3** PC with DEDUCE-DEP
1: **Input**: a set of variables **V**, CI tester
2: **Output**: a CPDAG

3: Initialize $\mathcal{G}$ with a complete undirected graph
4: **for** $k \in 1, 2, \ldots,$
5:    **for** an ordered pair of adjacent vertices $(X, Y) \in \mathcal{G}$ s.t. $|Ne(\{X\})_{\mathcal{G}} \setminus \{Y\}| \geq k$
6:       **for** $\mathbf{S} \subseteq Ne(\{X\})_{\mathcal{G}} \setminus \{Y\}$ s.t. $|\mathbf{S}| = k$
7:          **if** $(X \perp\!\!\!\perp Y \mid \mathbf{S})$
8:             **if not** DEDUCE-DEP$(\{X\}, \{Y\}, \mathbf{S})$
9:                Remove $X$-$Y$ from $\mathcal{G}$
10:      **else break**
11: Orient $\mathcal{G}$ for unshielded colliders
12: Complete orientation of $\mathcal{G}$ with Meek's rules
13: **return** $\mathcal{G}$

---

In Alg. 2, we augment HITON-PC-Nonsym with DEDUCE-DEP and illustrate how our method serves as a subroutine. Specifically, DEDUCE-DEP is called whenever a CIT outputs independence (lines 11–12). If DEDUCE-DEP confirms the CI statement, a subsequent removal operation is performed accordingly (line 13). It is worth noting that DEDUCE-DEP reuses previous low-order CIT results, enhancing the efficiency of correcting unreliable CITs.

### 4.2 PC WITH DEDUCE-DEP

PC algorithm [Spirtes et al., 2000] is a representative, sound and complete constraint-based method. It begins with a complete graph and proceeds to identify the skeleton of the underlying Bayesian network, subsequently orienting the edges. For skeleton identification, it conducts multiple CITs, removing an edge between variables $X$ and $Y$ if there exists a conditioning set $\mathbf{S}$ such that $(X \perp\!\!\!\perp Y \mid \mathbf{S})$. Since previous CIT results are used to orient other edges, even a single error from the unreliable CIT may incur other errors accordingly [Dash and Druzdzel, 2003, Tsamardinos et al., 2006]. Despite several attempts to mitigate error propagation [Colombo et al., 2014], this issue is still prevalent.

In Alg. 3, we integrate the DEDUCE-DEP into the PC algorithm. Analogous to the previous example, DEDUCE-DEP comes into play in the PC algorithm after performing CIT to confirm its result (line 8). If DEDUCE-DEP ultimately confirms the result of CIT, edges are removed (line 9). Considering that PC performs CITs by increasing the order of tests, our method will efficiently reuse many of the previous CIT results generated as the algorithm runs. In Sec. 5.3, we demonstrate that DEDUCE-DEP effectively controls the negative influence of *low-power* CITs, leading to improved performance for PC algorithm.[2]

---

[2]For the demonstration, we implemented PC-stable [Colombo et al., 2014], which is order-independent, making it more suitable for comparing the performance.

## 5 EMPIRICAL EVALUATIONS

We empirically investigate whether replacing the high-order CI statement with the result from deductive reasoning leads to a better performance in structure learning. Specifically, we first examine the effectiveness of our approach for correcting false negatives (FN) from CITs (Sec. 5.2). This allows us to assess the inherent capabilities and limitations of our method under controlled conditions. We then assess the performance improvement our method brings to constraint-based causal discovery algorithms (Sec. 5.3). This experiment enables us to assess the complementary nature of our method within the context of structure learning. We now describe the experimental setup.

### 5.1 EXPERIMENTAL SETUP

**Correcting CIT with DEDUCE-DEP.** In this experiment, we utilize random CI queries within random Bayesian networks (BNs) and compare the performance of our method with a counterpart where our method was not applied. This allows us to isolate the algorithm-dependent factors and purely validate the intrinsic effectiveness of our method: *how well does our method correct CI statements?* For this, we used the Erdös-Renyi model [Erdős et al., 1960] to generate random DAGs. Based on these DAG topologies, we designed conditional probability tables with binary variables, i.e., $P(V_i \mid Pa(V_i))$ follows Bernoulli distribution with its parameter uniformly sampled from $[0, 1]$. Following this process, we generated random BNs across 50 iterations, varying the number of variables ($|\mathbf{V}| \in \{10, 20, 30\}$) and the ratio of edges to variables ($|\mathbf{E}| / |\mathbf{V}| \in \{1.2, 1.5, 2\}$). For each random BN, data instances were sampled with the number of instances varying across $n \in \{200, 500, 1000\}$. Additionally, we composed random combinations of disjoint sets $X$, $Y$, and $\mathbf{Z}$ to formulate CI queries, repeating this process 20 times for each BN. The range of conditioning set sizes was set between 2 and 4.

**Causal discovery with DEDUCE-DEP.** We assess the performance improvement of HITON-PC with DEDUCE-DEP (Alg. 2) and PC with DEDUCE-DEP (Alg. 3). We used semi-synthetic datasets for evaluation. We sampled data instances from three benchmark BNs (i.e., Sachs, Alarm, and Insurance). HITON-PC and PC are both constraint-based; however, the former learns structure from an empty graph, whereas the latter learns structure from a complete graph. This experiment allows us to examine the efficacy of our method under different algorithmic behaviors. We note that we focused on the skeleton discovery phase from PC algorithm for evaluation.

**Implementation detail.** We employed the $G$-test for CIT to use. We set the default significance level for CIT to 0.05. The minimal conditioning set size $K$ is set to 1 for DEDUCE-DEP. We primarily utilized the following evaluation metrics:

Table 1: $|\mathbf{V}| = 20$ with 95% confidence interval. $N$ and $e$ denote the number of instances and edges, respectively.

| $N$ | Method | $e = 24$ | | | $e = 30$ | | | $e = 40$ | | |
|---|---|---|---|---|---|---|---|---|---|---|
| | | F1 | Precision | Recall | F1 | Precision | Recall | F1 | Precision | Recall |
| 200 | CIT | $0.26_{\pm 0.06}$ | $\mathbf{0.63}_{\pm 0.11}$ | $0.18_{\pm 0.04}$ | $0.28_{\pm 0.04}$ | $\mathbf{0.79}_{\pm 0.09}$ | $0.18_{\pm 0.03}$ | $0.30_{\pm 0.04}$ | $\mathbf{0.92}_{\pm 0.06}$ | $0.19_{\pm 0.03}$ |
| | CIT + DD | $\mathbf{0.59}_{\pm 0.03}$ | $0.50_{\pm 0.04}$ | $\mathbf{0.76}_{\pm 0.04}$ | $\mathbf{0.70}_{\pm 0.03}$ | $0.69_{\pm 0.04}$ | $\mathbf{0.73}_{\pm 0.03}$ | $\mathbf{0.78}_{\pm 0.03}$ | $0.86_{\pm 0.03}$ | $\mathbf{0.73}_{\pm 0.03}$ |
| 500 | CIT | $0.40_{\pm 0.05}$ | $\mathbf{0.78}_{\pm 0.08}$ | $0.29_{\pm 0.04}$ | $0.39_{\pm 0.05}$ | $\mathbf{0.84}_{\pm 0.07}$ | $0.26_{\pm 0.04}$ | $0.41_{\pm 0.03}$ | $\mathbf{0.97}_{\pm 0.02}$ | $0.26_{\pm 0.02}$ |
| | CIT + DD | $\mathbf{0.60}_{\pm 0.03}$ | $0.50_{\pm 0.04}$ | $\mathbf{0.78}_{\pm 0.04}$ | $\mathbf{0.71}_{\pm 0.04}$ | $0.66_{\pm 0.04}$ | $\mathbf{0.80}_{\pm 0.04}$ | $\mathbf{0.82}_{\pm 0.02}$ | $0.86_{\pm 0.03}$ | $\mathbf{0.78}_{\pm 0.03}$ |
| 1000 | CIT | $0.50_{\pm 0.05}$ | $\mathbf{0.86}_{\pm 0.05}$ | $0.37_{\pm 0.05}$ | $0.47_{\pm 0.04}$ | $\mathbf{0.89}_{\pm 0.06}$ | $0.33_{\pm 0.04}$ | $0.49_{\pm 0.04}$ | $\mathbf{0.97}_{\pm 0.02}$ | $0.34_{\pm 0.03}$ |
| | CIT + DD | $\mathbf{0.61}_{\pm 0.03}$ | $0.50_{\pm 0.04}$ | $\mathbf{0.81}_{\pm 0.04}$ | $\mathbf{0.74}_{\pm 0.02}$ | $0.70_{\pm 0.03}$ | $\mathbf{0.82}_{\pm 0.03}$ | $\mathbf{0.83}_{\pm 0.02}$ | $0.86_{\pm 0.03}$ | $\mathbf{0.81}_{\pm 0.03}$ |

precision, recall, and F1. We consider the output from the causal discovery algorithm with CI oracle as ground truth, and we compare our method's output to assess performance improvement. For HITON-PC, we calculated the evaluation score by averaging the score for every local skeleton learned. For PC, we calculated the evaluation score based on the global skeleton learned.

## 5.2 RESULTS OF CORRECTION EXPERIMENT

Table 1 presents the experimental results on random BN datasets with varying numbers of edges and data instances, where the number of variables is set to 20. It is evident that across all the cases listed, applying DEDUCE-DEP resulted in higher F1 scores compared to when it was not applied. While a significant increase in recall was accompanied by a corresponding decrease in precision, the overall improvement in F1 scores implies that our method does not indiscriminately retain any edge but rather accurately targets low-power tests to correct. However, a decrease in precision might indicate that the control of false positive rates might be important for further improving our method since statistical errors can also occur in CITs with dependence results. We also illustrate the results with similar experiments with the number of variables set to 10 and 30 in Figs. 9 to 11 in Appendix C.2. Furthermore, we present the experimental results on continuous data with linear/non-linear relationships Fig. 12 in Appendix C.2.

One interesting finding from Table 1 is that as the number of edges increases and the amount of data decreases, the difference in F1 scores between using and not using our method becomes more pronounced. Considering the experimental setup, CI queries randomly generated from dense graphs are more likely to indicate actual dependence. However, under such graph structures with a small amount of data, the original CIT tends to produce more false negatives due to the decrease in statistical power. In contrast, our method effectively corrects these false negatives, leading to an improvement in F1 scores. This suggests that our approach can be particularly useful in situations with low data volume and dense graph structures.

## 5.3 RESULTS OF PERFORMANCE EXPERIMENT

In the performance experiment, we investigate how much DEDUCE-DEP can enhance the overall performance of causal discovery algorithms by correcting unreliable CITs. As shown in Fig. 4 (values are reported in Tables 7 and 8 in Appendix C.2), DEDUCE-DEP consistently improves performance in terms of F1 score regardless of the algorithm used, with a significant increase in recall.

Notably, unlike the previous experiment, the precision when applying DEDUCE-DEP is quite comparable to cases where it is not applied. In cases where DEDUCE-DEP was applied, there were instances where both recall and precision increased compared to scenarios without it. This phenomenon can be explained by the interplay of false positives and false negatives [Armen and Tsamardinos, 2014]. False positives (in this context, wrongly added edges) occurring in the structure learning process can actually be induced by false negatives from previous steps. False positives, in turn, can incur false negatives (wrongly omitted edges), further propagating errors. Our experiment indicates that DEDUCE-DEP might help break this negative cycle to some extent.

**Evaluation after edge orientation.** So far, our focus has been primarily on the skeleton discovery phase of structure learning algorithms. This is because our method intervenes specifically in the skeleton learning step, while the subsequent steps remain unchanged for algorithms like PC (and HITON-PC focuses solely on finding neighbor sets for individual variables, leaving no further steps after skeleton discovery). To understand the broader impact of our method, we extend our analysis to orient edges after the skeleton learning step of PC and evaluate the resulting structure using the structural hamming distance (SHD).

As shown in Fig. 5, DEDUCE-DEP indeed leads to improvements in SHD as well, consistent with our previous findings in Fig. 4. This observation can be further illustrated through our motivating example in Sec. 1. Suppose the PC algorithm tries to learn the structure in Fig. 1. By correctly retaining the edge between $X$ and $Y$ through deductive reasoning, we can anticipate accurately identifying colliders, thereby facilitating the correct orientation of other edges. This indicates

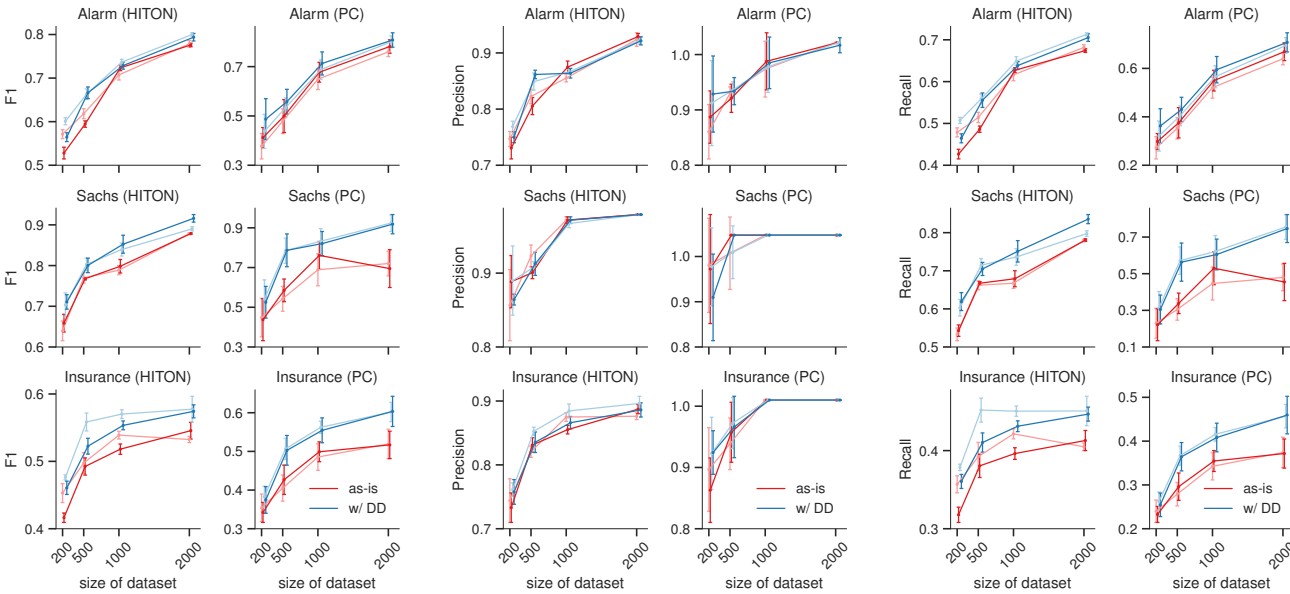

Figure 4: Performances (F1, precision, and recall) with varying dataset sizes. Blue (w/ DD) and red (as-is) lines are for with and without DEDUCE-DEP. Darker and lighter lines are for $\alpha = 0.05$ and $0.01$, respectively. Error bars are standard deviation.

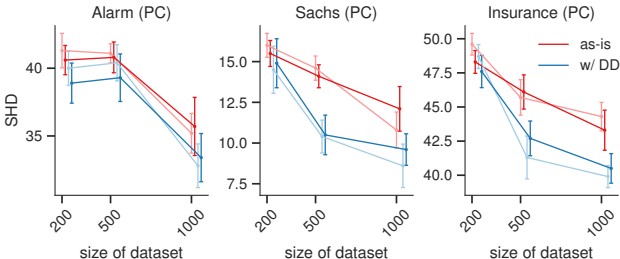

Figure 5: Results on real-world datasets measured in SHD. Darker and lighter lines are for $\alpha = 0.05$ and $0.01$, respectively. Error bars represent 95% confidence interval.

the importance of robust skeleton discovery in enhancing the performance of structure learning.

## 5.4 ANALYSIS

**Computational cost.** Applying DEDUCE-DEP to causal discovery algorithms may lead to an increase in the total number of CITs performed since it conducts new CITs if needed. In practice, our method efficiently reuses previously conducted CI information during the structure learning process. Fig. 6 demonstrates the computational performance measured in wall-clock time (seconds) and the number of CITs, across varying dataset sizes. We observe that computational costs are comparable to the original algorithm when data is scarce ($n = 200, 500, 1000$), indicating that our method efficiently corrects unreliable CI statements.

**Ablation on $K$.** We investigate the effect of the size of the

minimal conditioning set $K$ in DEDUCE-DEP. As shown in Fig. 7, a higher $K$ enhances computational efficiency by reducing the depth of recursion, whereas a lower $K$ tends to improve performance by allowing for more thorough exploration. Our method allows practitioners to tailor this trade-off between computational cost and performance to their specific needs and preferences.

**DEDUCE-DEP with reliability criterion.** We investigate how our method performs in conjunction with the reliability criterion heuristic. Specifically, we applied the heuristic power rule to the HITON-PC with a threshold of 5. As shown in Fig. 8, our method continues to improve performance over its counterpart. It is worth noting that this may not always be the case, as the reliability criterion heuristic distorts the whole structure learning process by selectively skipping certain CITs necessary for determining the underlying causal structure. We also observe only marginal performance improvement when data is scarce (e.g., $n = 200$). We speculate that this is because skipping CITs based on heuristics deprived our method of opportunities to correct tests, limiting its effectiveness.

## 6 RELATED WORKS

We categorize prior works addressing unreliable CITs into two main perspectives: namely, *internal* approaches focusing solely on the test itself and *external* approaches taking into account relationships with other tests.

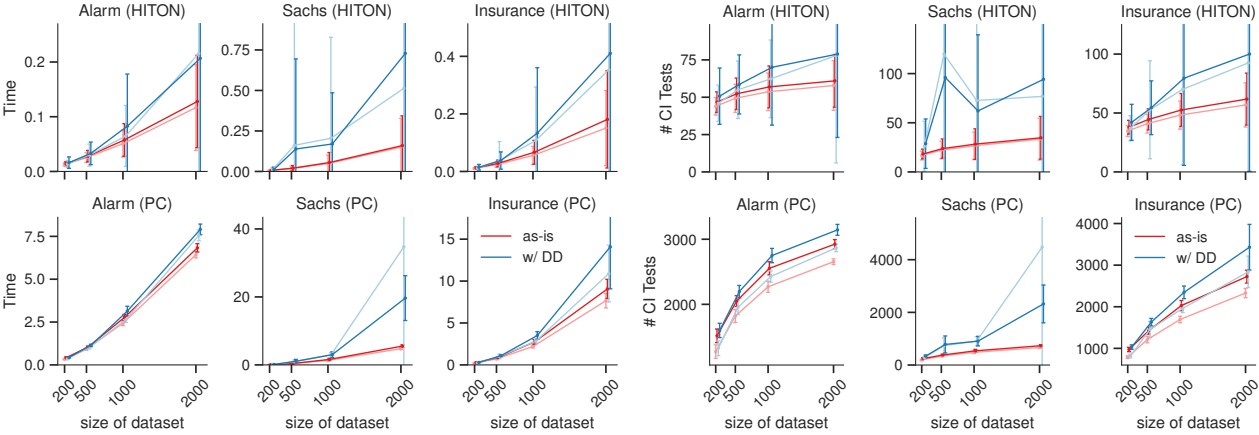

Figure 6: Computational performances (wall-clock time in seconds and the number of CITs) with varying dataset sizes. Blue and red lines are for with and without DEDUCE-DEP. Darker and lighter lines are for $\alpha = 0.05$ and $0.01$, respectively. Error bars are standard deviation.

## 6.1 INTERNAL APPROACHES

The majority of methods tackling reliability issues of CIT involves seeking solutions within the boundary of a single statistical test, such as enhancing the innate performance of the test or omitting the test itself. One prominent example is to employ heuristics for reliability criteria [Spirtes et al., 2000, Tsamardinos et al., 2006], which decides whether to perform or omit the test solely based on the amount of data needed. Although this renders simple measures to tackle the unreliable CITs, this lacks theoretical soundness, and to our knowledge, it is only applicable to discrete data.

Another example involves opting for permutation-based tests over classical asymptotic ones (e.g., $\chi^2$ test, $G$-test) [Tsamardinos and Borboudakis, 2010]. As permutation-based tests exhibit better calibration under data-scarce scenarios, various works extend permutation-based CIT to tackle various regimes (e.g., continuous settings), with the prospect of broadly applying it to structure learning [Doran et al., 2014, Lee and Honavar, 2017, Berrett et al., 2020, Zhang et al., 2022, Kim et al., 2023].

Recent research suggested causal structure learning within the confines of low-order CITs [Wienöbst and Liskiewicz, 2020, Kocaoglu, 2024]. By excluding unreliable, high-order CITs, they attempt to yield more robust causal structures. However, such approaches may lack the full specification of a graph structure.

## 6.2 EXTERNAL APPROACHES

Another line of work goes beyond individual tests and considers relationships with other tests, addressing statistical errors of CITs with conflict resolution among inconsistent CI statements. By translating CI statements from data into logical constraints, researchers aim to search for a causal

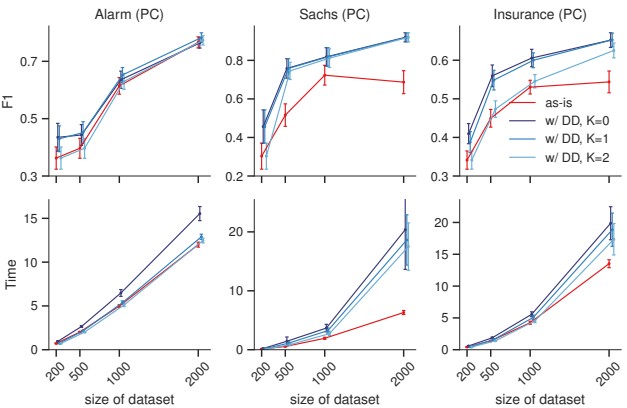

Figure 7: Ablation on $K$. (**Top**) F1 score. (**Bottom**) Computational performances measured as wall-clock time (seconds). Error bars represent 95% confidence interval.

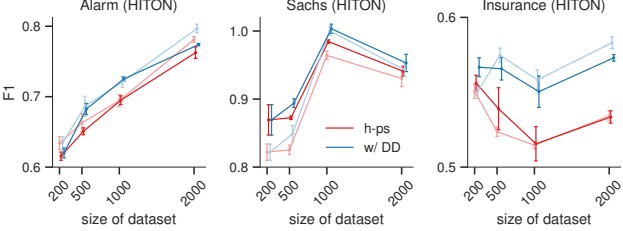

Figure 8: Performances with the reliability criterion (heuristic power rule) applied, measured in F1 score. Darker and lighter lines are for $\alpha = 0.05$ and $0.01$, respectively. Error bars represent 95% confidence interval.

structure consistent with all the encoded constraints using various solvers (e.g., SAT, ASP solvers) [Hyttinen et al., 2013, 2014]. Another prominent example is utilizing rules from graphoid axioms [Bromberg and Margaritis, 2009, Ma et al., 2023]. This involves adopting either a preference-based argumentative framework [Bromberg and Margaritis, 2009] or off-the-shelf SMT solver [Ma et al., 2023], designed to facilitate reasoning with CI statements. However, they often involve complicated routines and high computational costs, limiting their practical applicability.

## 6.3 DISCUSSION

In contrast to prior approaches, our work combines internal and external perspectives, considering the relationships with other CITs but strictly limited to lower-order ones that are deemed reliable. This strategy allows DEDUCE-DEP to selectively apply rules derived from graphoid axioms, providing a more efficient and principled means of constraining higher-order CIT results and avoiding the complicated computation associated with all possible combinations of rules from graphoid axioms.

Additionally, DEDUCE-DEP actively executes new CITs, unlike other methods utilizing graphoid axioms. This allows us to effectively correct unreliable CITs. For instance, consider the scenario where a structure learning algorithm attempts to learn a causal graph in Fig. 1. Initially, the algorithm performs unconditional independence tests, yielding $(Y \perp\!\!\!\perp Z'')$, which we assume to be correct. Subsequently, it no longer examines CI between $Y$ and $Z''$ given any conditioning set, resulting in $(Y \perp\!\!\!\perp Z'' \mid Z')$, not available. In such a case, inferring higher-order CI statements solely based on previously performed lower-order CIT results might be limited, potentially failing to correct $(X \perp\!\!\!\perp Y \mid Z', Z'')$. Without performing additional tests, like $(Y \perp\!\!\!\perp Z'' \mid Z')$, the correction of unreliable CITs can be restricted. This suggests that correcting false negatives might require more than just resolving conflicts among existing CIT results.

## 7 CONCLUSION

We presented DEDUCE-DEP, a simple, principled, and practical correction method for addressing unreliable CITs using deductive reasoning. By leveraging rules derived from graphoid axioms, we explored the conditions for deducing high-order CI statements from low-order CI statements and integrated these rules into our algorithm. DEDUCE-DEP systematically replaces unreliable independence statements with deductively reasoned dependence statements from lower-order CITs. We showed how our method can be seamlessly integrated into causal discovery algorithms like HITON-PC or PC and provided empirical evidence of its efficacy. Despite its distinct results, there still remains a need

to address dependence statements to prevent errors from false positives. Therefore, future research directions may involve combining our method with existing false positive control methods [Li and Wang, 2009, Strobl et al., 2019] for a more robust causal discovery. Another promising future direction is extending our framework to local independence relationships, e.g., context-specific independence [Boutilier et al., 1996, Pensar et al., 2016, Hwang et al., 2023].

## Acknowledgements

We would like to thank anonymous reviewers for their constructive comments. This work was partly supported by the IITP (2022-0-00953-PICA/30%) and NRF (RS-2023-00222663/30%, RS-2023-00211904/40%) grant funded by the Korean government.

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

# Causal Discovery with Deductive Reasoning: One Less Problem (Supplementary Material)

**Jonghwan Kim**[1]      **Inwoo Hwang**[2]      **Sanghack Lee**[* 1,2]

[1]Graduate School of Data Science, Seoul National University, South Korea
[2]Artificial Intelligence Institute, Seoul National University, South Korea
[*]Correspondence to: sanghack@snu.ac.kr

## A   RULES DERIVED FROM GRAPHOID AXIOMS

We present rules derived from graphoid axioms [Pearl and Paz, 1987, Geiger, 1990, Bromberg and Margaritis, 2009].

$$\text{(Symmetry) } (\mathbf{X} \perp\!\!\!\perp \mathbf{Y} \mid \mathbf{Z}) \iff (\mathbf{Y} \perp\!\!\!\perp \mathbf{X} \mid \mathbf{Z})$$
$$\text{(Decomposition) } (\mathbf{X} \perp\!\!\!\perp \mathbf{Y}, \mathbf{W} \mid \mathbf{Z}) \implies (\mathbf{X} \perp\!\!\!\perp \mathbf{Y} \mid \mathbf{Z}) \wedge (\mathbf{X} \perp\!\!\!\perp \mathbf{W} \mid \mathbf{Z})$$
$$\text{(Weak Union) } (\mathbf{X} \perp\!\!\!\perp \mathbf{Y}, \mathbf{W} \mid \mathbf{Z}) \implies (\mathbf{X} \perp\!\!\!\perp \mathbf{Y} \mid \mathbf{Z}, \mathbf{W})$$
$$\text{(Contraction) } (\mathbf{X} \perp\!\!\!\perp \mathbf{Y} \mid \mathbf{Z}) \wedge (\mathbf{X} \perp\!\!\!\perp \mathbf{W} \mid \mathbf{Z}, \mathbf{Y}) \implies (\mathbf{X} \perp\!\!\!\perp \mathbf{Y}, \mathbf{W} \mid \mathbf{Z})$$
$$\text{(Intersection) } (\mathbf{X} \perp\!\!\!\perp \mathbf{Y} \mid \mathbf{Z}, \mathbf{W}) \wedge (\mathbf{X} \perp\!\!\!\perp \mathbf{W} \mid \mathbf{Z}, \mathbf{Y}) \implies (\mathbf{X} \perp\!\!\!\perp \mathbf{Y}, \mathbf{W} \mid \mathbf{Z})$$

Under a (causal) faithfulness assumption, we have a more relaxed set of rules as follows.

$$\text{(Symmetry) } (\mathbf{X} \perp\!\!\!\perp \mathbf{Y} \mid \mathbf{Z}) \iff (\mathbf{Y} \perp\!\!\!\perp \mathbf{X} \mid \mathbf{Z})$$
$$\text{(Composition) } (\mathbf{X} \perp\!\!\!\perp \mathbf{Y} \mid \mathbf{Z}) \wedge (\mathbf{X} \perp\!\!\!\perp \mathbf{W} \mid \mathbf{Z}) \implies (\mathbf{X} \perp\!\!\!\perp \mathbf{Y}, \mathbf{W} \mid \mathbf{Z})$$
$$\text{(Decomposition) } (\mathbf{X} \perp\!\!\!\perp \mathbf{Y}, \mathbf{W} \mid \mathbf{Z}) \implies (\mathbf{X} \perp\!\!\!\perp \mathbf{Y} \mid \mathbf{Z}) \wedge (\mathbf{X} \perp\!\!\!\perp \mathbf{W} \mid \mathbf{Z})$$
$$\text{(Intersection) } (\mathbf{X} \perp\!\!\!\perp \mathbf{Y} \mid \mathbf{Z}, \mathbf{W}) \wedge (\mathbf{X} \perp\!\!\!\perp \mathbf{W} \mid \mathbf{Z}, \mathbf{Y}) \implies (\mathbf{X} \perp\!\!\!\perp \mathbf{Y}, \mathbf{W} \mid \mathbf{Z})$$
$$\text{(Weak Union) } (\mathbf{X} \perp\!\!\!\perp \mathbf{Y}, \mathbf{W} \mid \mathbf{Z}) \implies (\mathbf{X} \perp\!\!\!\perp \mathbf{Y} \mid \mathbf{Z}, \mathbf{W})$$
$$\text{(Contraction) } (\mathbf{X} \perp\!\!\!\perp \mathbf{Y} \mid \mathbf{Z}) \wedge (\mathbf{X} \perp\!\!\!\perp \mathbf{W} \mid \mathbf{Z}, \mathbf{Y}) \implies (\mathbf{X} \perp\!\!\!\perp \mathbf{Y}, \mathbf{W} \mid \mathbf{Z})$$
$$\text{(Weak Transitivity) } (\mathbf{X} \perp\!\!\!\perp \mathbf{Y} \mid \mathbf{Z}) \wedge (\mathbf{X} \perp\!\!\!\perp \mathbf{Y} \mid \mathbf{Z}, W) \implies (\mathbf{X} \perp\!\!\!\perp W \mid \mathbf{Z}) \vee (W \perp\!\!\!\perp \mathbf{Y} \mid \mathbf{Z})$$
$$\text{(Chordality) } (X \perp\!\!\!\perp Y \mid W, Z) \wedge (W \perp\!\!\!\perp Z \mid X, Y) \implies (X \perp\!\!\!\perp Y \mid W) \vee (X \perp\!\!\!\perp Y \mid Z)$$

Each non-bold letter in weak transitivity and chordality is a variable (i.e., a singleton).

## B   OMITTED PROOFS

**Proposition 1.** *Under the faithful Bayesian network* $(\mathcal{G}, P)$*, let* $\mathbf{X}$*,* $\mathbf{Y}$*, and* $\mathbf{Z}$ *be disjoint subsets of* $\mathbf{V}$ *where* $\mathbf{Z}$ *is partitioned into* $\mathbf{Z}'$ *and* $\mathbf{Z}''$ *such that* $\mathbf{Z} = \mathbf{Z}' \sqcup \mathbf{Z}'', |\mathbf{Z}''| = 1$*. Then,* $(\mathbf{X} \not\perp\!\!\!\perp \mathbf{Y} \mid \mathbf{Z})$ *if one of the following holds:*

1. $(\mathbf{X} \not\perp\!\!\!\perp \mathbf{Y} \mid \mathbf{Z}') \wedge (\mathbf{X} \perp\!\!\!\perp \mathbf{Z}'' \mid \mathbf{Z}')$
2. $(\mathbf{X} \perp\!\!\!\perp \mathbf{Y} \mid \mathbf{Z}') \wedge (\mathbf{X} \not\perp\!\!\!\perp \mathbf{Z}'' \mid \mathbf{Z}') \wedge (\mathbf{Y} \not\perp\!\!\!\perp \mathbf{Z}'' \mid \mathbf{Z}')$

*Proof.* We prove each item below.

(1) We begin with the definition of the contraction rule:

$$(\mathbf{X} \perp\!\!\!\perp \mathbf{Z}'' \mid \mathbf{Z}') \wedge (\mathbf{X} \perp\!\!\!\perp \mathbf{Y} \mid \mathbf{Z}', \mathbf{Z}'')$$
$$\implies (\mathbf{X} \perp\!\!\!\perp \mathbf{Y}, \mathbf{Z}'' \mid \mathbf{Z}').$$

By the decomposition rule, we have:

$$\implies (\mathbf{X} \perp\!\!\!\perp \mathbf{Z}'' \mid \mathbf{Z}') \wedge (\mathbf{X} \perp\!\!\!\perp \mathbf{Y} \mid \mathbf{Z}').$$

Now, by taking contraposition, we acquire

$$(\mathbf{X} \not\perp\!\!\!\perp \mathbf{Z}'' \mid \mathbf{Z}') \vee (\mathbf{X} \not\perp\!\!\!\perp \mathbf{Y} \mid \mathbf{Z}') \implies$$
$$(\mathbf{X} \not\perp\!\!\!\perp \mathbf{Z}'' \mid \mathbf{Z}') \vee (\mathbf{X} \not\perp\!\!\!\perp \mathbf{Y} \mid \mathbf{Z}', \mathbf{Z}'').$$

Therefore, if $(\mathbf{X} \not\perp\!\!\!\perp \mathbf{Y} \mid \mathbf{Z}')$ and $(\mathbf{X} \perp\!\!\!\perp \mathbf{Z}'' \mid \mathbf{Z}')$, then $(\mathbf{X} \not\perp\!\!\!\perp \mathbf{Y} \mid \mathbf{Z}' \cup \mathbf{Z}'')$.

(2) By weak transitivity rule,

$$(\mathbf{X} \perp\!\!\!\perp \mathbf{Y} \mid \mathbf{Z}') \wedge (\mathbf{X} \perp\!\!\!\perp \mathbf{Y} \mid \mathbf{Z}', \mathbf{Z}'') \implies$$
$$(\mathbf{X} \perp\!\!\!\perp \mathbf{Z}'' \mid \mathbf{Z}') \vee (\mathbf{Y} \perp\!\!\!\perp \mathbf{Z}'' \mid \mathbf{Z}').$$

By contraposition, $(\mathbf{X} \not\perp\!\!\!\perp \mathbf{Z}'' \mid \mathbf{Z}') \wedge (\mathbf{Y} \not\perp\!\!\!\perp \mathbf{Z}'' \mid \mathbf{Z}') \implies (\mathbf{X} \not\perp\!\!\!\perp \mathbf{Y} \mid \mathbf{Z}') \vee (\mathbf{X} \not\perp\!\!\!\perp \mathbf{Y} \mid \mathbf{Z}', \mathbf{Z}'')$. Therefore, if

$$(\mathbf{X} \perp\!\!\!\perp \mathbf{Y} \mid \mathbf{Z}') \wedge (\mathbf{X} \not\perp\!\!\!\perp \mathbf{Z}'' \mid \mathbf{Z}') \wedge (\mathbf{Y} \not\perp\!\!\!\perp \mathbf{Z}'' \mid \mathbf{Z}'),$$

then $(\mathbf{X} \not\perp\!\!\!\perp \mathbf{Y} \mid \mathbf{Z}', \mathbf{Z}'')$. $\qquad\square$

# C  APPENDIX FOR EXPERIMENTS

## C.1  EXPERIMENTAL DETAILS

### C.1.1  Datasets

Sachs dataset [Sachs et al., 2005] pertains to protein expression in human immune system cells. It encompasses simultaneous measurements of 11 phosphorylated proteins and phospholipids obtained from thousands of individual primary immune system cells. The dataset comprises 11 vertices representing the different proteins and phospholipids, with a total of 17 edges indicating the relationships between them. On average, each node has a degree of 3.09, with the maximum in-degree being 3.

ALARM dataset [Beinlich et al., 1989], short for "A Logical Alarm Reduction Mechanism," represents a Bayesian network tailored to serve as an alarm message system for patient monitoring. It consists of 37 variables, each representing various factors of patient health and monitoring parameters. These nodes are interconnected by 46 edges, reflecting the relationships between different variables in the network. On average, each node has a degree of 2.49, with the maximum in-degree being 4.

Insurance dataset [Binder et al., 1997] is designed for evaluating car insurance risks. It comprises 27 variables representing different features related to insurance risk assessment. These variables are associated with each other by 52 edges. On average, each node in the Insurance network has a degree of 3.85, with the maximum in-degree being 3.

### C.1.2  Implementation details

We utilized several semi-synthetic BN datasets from the repository of R package BNlearn [Scutari, 2010]. All experiments were processed using Intel(R) Xeon(R) Gold 6342 CPU @ 2.80GHz. Our code is available at https://github.com/snu-causality-lab/deduce-dep.

## C.2  ADDITIONAL EXPERIMENTAL RESULTS

We provide all experimental results with varying numbers of nodes ($|\mathbf{V}| = 10, 20, 30$) in Figs. 9 to 11 and Tables 2 to 8.

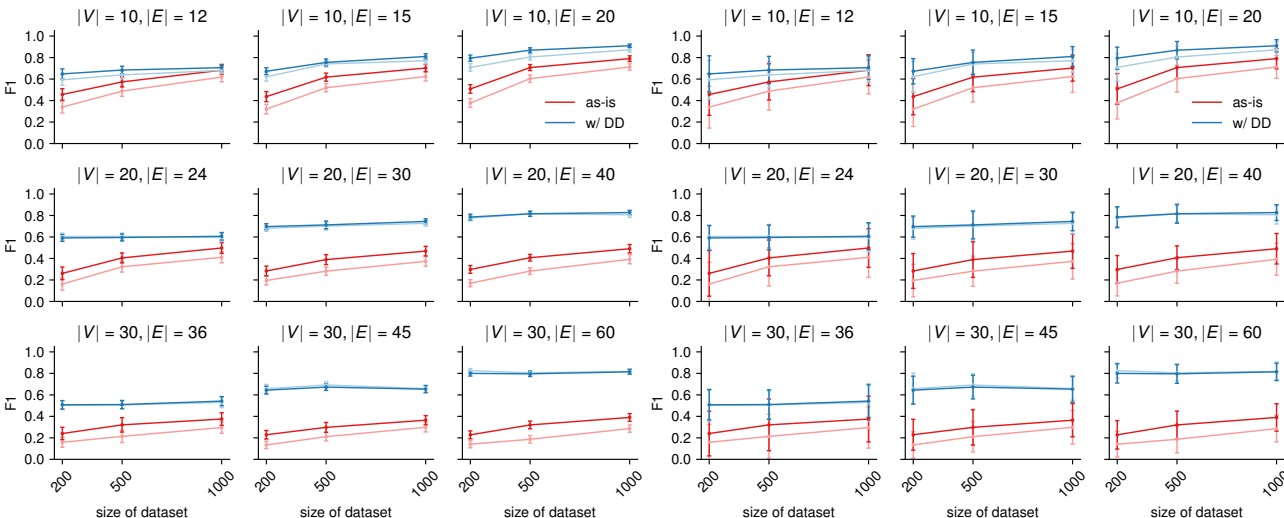

Figure 9: Results of Correction Experiment (F1) (Left, 95% Confidence Interval, Right, Standard Deviation).

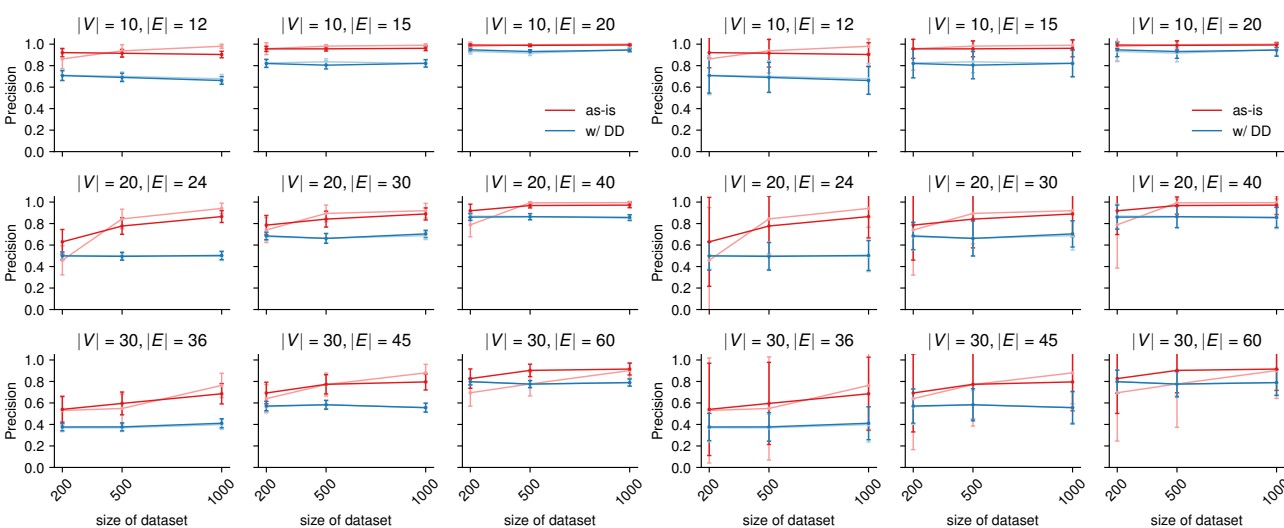

Figure 10: Results of Correction Experiment (Precision) (Left, 95% Confidence Interval, Right, Standard Deviation).

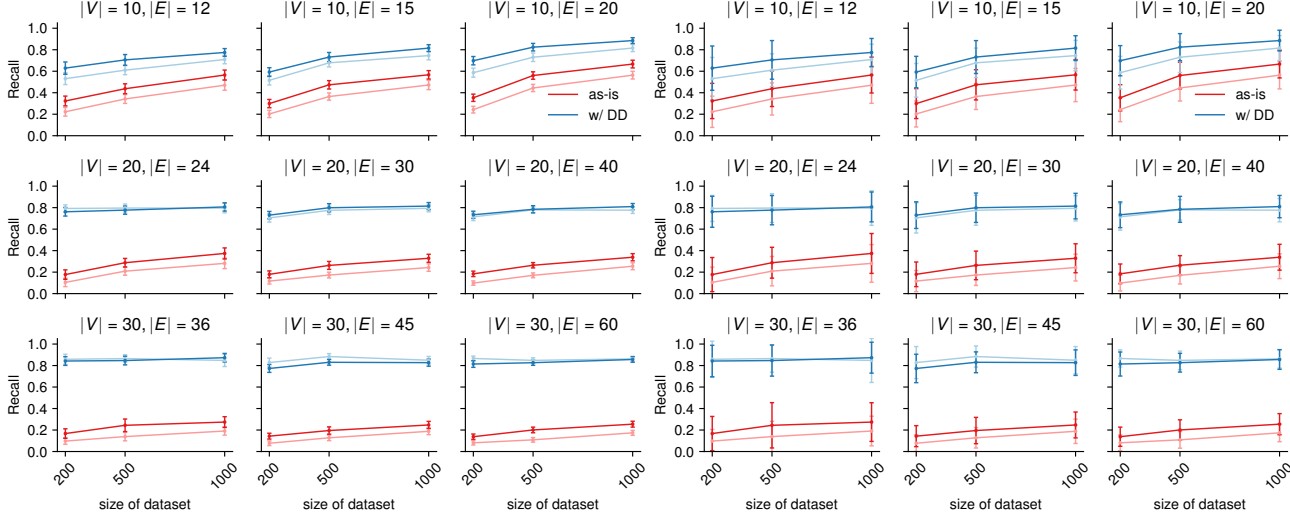

Figure 11: Results of Correction Experiment (Recall) (Left, 95% Confidence Interval, Right, Standard Deviation).

Table 2: $|\mathbf{V}| = 10$ with Standard Deviation.

| N | Method | e = 12 | | | e = 15 | | | e = 20 | | |
|---|---|---|---|---|---|---|---|---|---|---|
| | | F1 | Precision | Recall | F1 | Precision | Recall | F1 | Precision | Recall |
| 200 | CIT | $0.46_{\pm 0.19}$ | $0.92_{\pm 0.14}$ | $0.32_{\pm 0.16}$ | $0.44_{\pm 0.17}$ | $0.96_{\pm 0.09}$ | $0.30_{\pm 0.14}$ | $0.51_{\pm 0.14}$ | $0.99_{\pm 0.03}$ | $0.35_{\pm 0.12}$ |
| | CIT + DD | $0.65_{\pm 0.17}$ | $0.71_{\pm 0.16}$ | $0.63_{\pm 0.21}$ | $0.67_{\pm 0.12}$ | $0.82_{\pm 0.13}$ | $0.59_{\pm 0.15}$ | $0.79_{\pm 0.10}$ | $0.95_{\pm 0.06}$ | $0.70_{\pm 0.14}$ |
| 500 | CIT | $0.57_{\pm 0.17}$ | $0.92_{\pm 0.13}$ | $0.44_{\pm 0.17}$ | $0.62_{\pm 0.14}$ | $0.96_{\pm 0.07}$ | $0.47_{\pm 0.14}$ | $0.71_{\pm 0.10}$ | $0.99_{\pm 0.04}$ | $0.56_{\pm 0.12}$ |
| | CIT + DD | $0.68_{\pm 0.13}$ | $0.69_{\pm 0.14}$ | $0.71_{\pm 0.18}$ | $0.76_{\pm 0.11}$ | $0.8_{\pm 0.13}$ | $0.73_{\pm 0.15}$ | $0.87_{\pm 0.08}$ | $0.93_{\pm 0.06}$ | $0.82_{\pm 0.13}$ |
| 1000 | CIT | $0.68_{\pm 0.14}$ | $0.90_{\pm 0.11}$ | $0.56_{\pm 0.17}$ | $0.70_{\pm 0.12}$ | $0.96_{\pm 0.08}$ | $0.57_{\pm 0.14}$ | $0.79_{\pm 0.09}$ | $0.99_{\pm 0.02}$ | $0.67_{\pm 0.13}$ |
| | CIT + DD | $0.71_{\pm 0.11}$ | $0.66_{\pm 0.13}$ | $0.78_{\pm 0.13}$ | $0.81_{\pm 0.09}$ | $0.82_{\pm 0.12}$ | $0.82_{\pm 0.12}$ | $0.91_{\pm 0.06}$ | $0.94_{\pm 0.06}$ | $0.88_{\pm 0.10}$ |

Table 3: $|\mathbf{V}| = 10$ with 95% Confidence Interval.

| N | Method | e = 12 | | | e = 15 | | | e = 20 | | |
|---|---|---|---|---|---|---|---|---|---|---|
| | | F1 | Precision | Recall | F1 | Precision | Recall | F1 | Precision | Recall |
| 200 | CIT | $0.46_{\pm 0.05}$ | $0.92_{\pm 0.04}$ | $0.32_{\pm 0.05}$ | $0.44_{\pm 0.05}$ | $0.96_{\pm 0.02}$ | $0.30_{\pm 0.04}$ | $0.51_{\pm 0.04}$ | $0.99_{\pm 0.01}$ | $0.35_{\pm 0.03}$ |
| | CIT + DD | $0.65_{\pm 0.05}$ | $0.71_{\pm 0.04}$ | $0.63_{\pm 0.06}$ | $0.67_{\pm 0.03}$ | $0.82_{\pm 0.04}$ | $0.59_{\pm 0.04}$ | $0.79_{\pm 0.03}$ | $0.95_{\pm 0.02}$ | $0.70_{\pm 0.04}$ |
| 500 | CIT | $0.57_{\pm 0.05}$ | $0.92_{\pm 0.04}$ | $0.44_{\pm 0.05}$ | $0.62_{\pm 0.04}$ | $0.96_{\pm 0.02}$ | $0.47_{\pm 0.04}$ | $0.71_{\pm 0.03}$ | $0.99_{\pm 0.01}$ | $0.56_{\pm 0.03}$ |
| | CIT + DD | $0.68_{\pm 0.03}$ | $0.69_{\pm 0.04}$ | $0.71_{\pm 0.05}$ | $0.76_{\pm 0.03}$ | $0.80_{\pm 0.03}$ | $0.73_{\pm 0.04}$ | $0.87_{\pm 0.02}$ | $0.93_{\pm 0.02}$ | $0.82_{\pm 0.03}$ |
| 1000 | CIT | $0.68_{\pm 0.04}$ | $0.90_{\pm 0.03}$ | $0.56_{\pm 0.05}$ | $0.70_{\pm 0.03}$ | $0.96_{\pm 0.02}$ | $0.57_{\pm 0.04}$ | $0.79_{\pm 0.03}$ | $0.99_{\pm 0.01}$ | $0.67_{\pm 0.04}$ |
| | CIT + DD | $0.71_{\pm 0.03}$ | $0.66_{\pm 0.04}$ | $0.78_{\pm 0.04}$ | $0.81_{\pm 0.03}$ | $0.82_{\pm 0.03}$ | $0.82_{\pm 0.03}$ | $0.91_{\pm 0.02}$ | $0.94_{\pm 0.02}$ | $0.88_{\pm 0.03}$ |

Table 4: $|\mathbf{V}| = 20$ with Standard Deviation.

| N | Method | e = 24 | | | e = 30 | | | e = 40 | | |
|---|---|---|---|---|---|---|---|---|---|---|
| | | F1 | Precision | Recall | F1 | Precision | Recall | F1 | Precision | Recall |
| 200 | CIT | $0.26_{\pm 0.21}$ | $0.63_{\pm 0.41}$ | $0.18_{\pm 0.16}$ | $0.28_{\pm 0.16}$ | $0.79_{\pm 0.33}$ | $0.18_{\pm 0.11}$ | $0.30_{\pm 0.13}$ | $0.92_{\pm 0.22}$ | $0.19_{\pm 0.09}$ |
| | CIT + DD | $0.59_{\pm 0.11}$ | $0.50_{\pm 0.13}$ | $0.76_{\pm 0.14}$ | $0.70_{\pm 0.10}$ | $0.69_{\pm 0.13}$ | $0.73_{\pm 0.12}$ | $0.78_{\pm 0.10}$ | $0.86_{\pm 0.11}$ | $0.73_{\pm 0.12}$ |
| 500 | CIT | $0.40_{\pm 0.17}$ | $0.78_{\pm 0.28}$ | $0.29_{\pm 0.14}$ | $0.39_{\pm 0.17}$ | $0.84_{\pm 0.27}$ | $0.26_{\pm 0.13}$ | $0.41_{\pm 0.11}$ | $0.97_{\pm 0.08}$ | $0.26_{\pm 0.09}$ |
| | CIT + DD | $0.60_{\pm 0.12}$ | $0.50_{\pm 0.13}$ | $0.78_{\pm 0.14}$ | $0.71_{\pm 0.13}$ | $0.66_{\pm 0.16}$ | $0.80_{\pm 0.14}$ | $0.82_{\pm 0.09}$ | $0.86_{\pm 0.1}$ | $0.78_{\pm 0.12}$ |
| 1000 | CIT | $0.50_{\pm 0.18}$ | $0.86_{\pm 0.20}$ | $0.37_{\pm 0.19}$ | $0.47_{\pm 0.16}$ | $0.89_{\pm 0.2}$ | $0.33_{\pm 0.13}$ | $0.49_{\pm 0.14}$ | $0.97_{\pm 0.09}$ | $0.34_{\pm 0.12}$ |
| | CIT + DD | $0.61_{\pm 0.13}$ | $0.50_{\pm 0.14}$ | $0.81_{\pm 0.14}$ | $0.74_{\pm 0.09}$ | $0.70_{\pm 0.12}$ | $0.82_{\pm 0.12}$ | $0.83_{\pm 0.07}$ | $0.86_{\pm 0.09}$ | $0.81_{\pm 0.10}$ |

Table 5: $|\mathbf{V}| = 30$ with Standard Deviation.

| N | Method | e = 36 | | | e = 45 | | | e = 60 | | |
|---|---|---|---|---|---|---|---|---|---|---|
| | | F1 | Precision | Recall | F1 | Precision | Recall | F1 | Precision | Recall |
| 200 | CIT | $0.24_{\pm 0.21}$ | $0.54_{\pm 0.43}$ | $0.17_{\pm 0.16}$ | $0.23_{\pm 0.14}$ | $0.69_{\pm 0.36}$ | $0.14_{\pm 0.10}$ | $0.23_{\pm 0.13}$ | $0.83_{\pm 0.32}$ | $0.14_{\pm 0.09}$ |
| | CIT + DD | $0.51_{\pm 0.14}$ | $0.38_{\pm 0.13}$ | $0.84_{\pm 0.15}$ | $0.64_{\pm 0.13}$ | $0.57_{\pm 0.16}$ | $0.77_{\pm 0.13}$ | $0.8_{\pm 0.09}$ | $0.80_{\pm 0.11}$ | $0.81_{\pm 0.11}$ |
| 500 | CIT | $0.32_{\pm 0.24}$ | $0.60_{\pm 0.38}$ | $0.24_{\pm 0.21}$ | $0.30_{\pm 0.17}$ | $0.77_{\pm 0.32}$ | $0.19_{\pm 0.12}$ | $0.32_{\pm 0.13}$ | $0.90_{\pm 0.21}$ | $0.20_{\pm 0.10}$ |
| | CIT + DD | $0.51_{\pm 0.14}$ | $0.38_{\pm 0.13}$ | $0.85_{\pm 0.14}$ | $0.67_{\pm 0.11}$ | $0.58_{\pm 0.15}$ | $0.83_{\pm 0.10}$ | $0.80_{\pm 0.09}$ | $0.78_{\pm 0.12}$ | $0.83_{\pm 0.09}$ |
| 1000 | CIT | $0.38_{\pm 0.21}$ | $0.69_{\pm 0.34}$ | $0.27_{\pm 0.18}$ | $0.36_{\pm 0.15}$ | $0.80_{\pm 0.27}$ | $0.25_{\pm 0.12}$ | $0.39_{\pm 0.13}$ | $0.92_{\pm 0.20}$ | $0.25_{\pm 0.10}$ |
| | CIT + DD | $0.54_{\pm 0.15}$ | $0.41_{\pm 0.15}$ | $0.87_{\pm 0.14}$ | $0.65_{\pm 0.12}$ | $0.56_{\pm 0.15}$ | $0.83_{\pm 0.12}$ | $0.81_{\pm 0.08}$ | $0.79_{\pm 0.12}$ | $0.86_{\pm 0.09}$ |

Table 6: $|\mathbf{V}| = 30$ with 95% Confidence Interval.

| N | Method | e = 36 | | | e = 45 | | | e = 60 | | |
|---|---|---|---|---|---|---|---|---|---|---|
| | | F1 | Precision | Recall | F1 | Precision | Recall | F1 | Precision | Recall |
| 200 | CIT | 0.24±0.06 | 0.54±0.12 | 0.17±0.04 | 0.23±0.04 | 0.69±0.10 | 0.14±0.03 | 0.23±0.04 | 0.83±0.09 | 0.14±0.02 |
| | CIT + DD | 0.51±0.04 | 0.38±0.03 | 0.84±0.04 | 0.64±0.04 | 0.57±0.04 | 0.77±0.04 | 0.80±0.02 | 0.80±0.03 | 0.81±0.03 |
| 500 | CIT | 0.32±0.07 | 0.60±0.10 | 0.24±0.06 | 0.30±0.05 | 0.77±0.09 | 0.19±0.03 | 0.32±0.04 | 0.90±0.06 | 0.20±0.03 |
| | CIT + DD | 0.51±0.04 | 0.38±0.04 | 0.85±0.04 | 0.67±0.03 | 0.58±0.04 | 0.83±0.03 | 0.80±0.02 | 0.78±0.03 | 0.83±0.02 |
| 1000 | CIT | 0.38±0.06 | 0.69±0.09 | 0.27±0.05 | 0.36±0.04 | 0.80±0.07 | 0.25±0.03 | 0.39±0.03 | 0.92±0.05 | 0.25±0.03 |
| | CIT + DD | 0.54±0.04 | 0.41±0.04 | 0.87±0.04 | 0.65±0.03 | 0.56±0.04 | 0.83±0.03 | 0.81±0.02 | 0.79±0.03 | 0.86±0.03 |

Table 7: Results of Performance Experiment (PC).

| N | Method | Alarm | | | Sachs | | | Insurance | | |
|---|---|---|---|---|---|---|---|---|---|---|
| | | F1 | Precision | Recall | F1 | Precision | Recall | F1 | Precision | Recall |
| 200 | PC | 0.41±0.04 | 0.87±0.05 | 0.27±0.03 | 0.40±0.11 | 0.92±0.12 | 0.26±0.09 | 0.37±0.03 | 0.85±0.05 | 0.24±0.02 |
| | PC + DD | 0.49±0.08 | 0.91±0.07 | 0.34±0.07 | 0.48±0.08 | 0.86±0.10 | 0.34±0.08 | 0.41±0.03 | 0.91±0.04 | 0.26±0.03 |
| 500 | PC | 0.50±0.07 | 0.90±0.03 | 0.35±0.06 | 0.54±0.06 | 1.00±0.00 | 0.37±0.06 | 0.46±0.04 | 0.95±0.05 | 0.30±0.03 |
| | PC + DD | 0.56±0.05 | 0.91±0.02 | 0.40±0.05 | 0.74±0.08 | 1.00±0.00 | 0.60±0.10 | 0.53±0.04 | 0.96±0.05 | 0.37±0.03 |
| 1000 | PC | 0.68±0.04 | 0.97±0.05 | 0.52±0.04 | 0.72±0.07 | 1.00±0.00 | 0.57±0.09 | 0.53±0.03 | 1.00±0.00 | 0.36±0.02 |
| | PC + DD | 0.71±0.05 | 0.96±0.05 | 0.57±0.05 | 0.78±0.06 | 1.00±0.00 | 0.64±0.08 | 0.59±0.03 | 1.00±0.00 | 0.41±0.03 |
| 2000 | PC | 0.78±0.03 | 1.00±0.00 | 0.64±0.04 | 0.65±0.10 | 1.00±0.00 | 0.49±0.10 | 0.55±0.04 | 1.00±0.00 | 0.38±0.03 |
| | PC + DD | 0.81±0.03 | 1.00±0.01 | 0.68±0.04 | 0.88±0.05 | 1.00±0.00 | 0.78±0.08 | 0.63±0.04 | 1.00±0.00 | 0.47±0.04 |

Table 8: Results of Performance Experiment (HITON-PC).

| N | Method | Alarm | | | Sachs | | | Insurance | | |
|---|---|---|---|---|---|---|---|---|---|---|
| | | F1 | Precision | Recall | F1 | Precision | Recall | F1 | Precision | Recall |
| 200 | HITON | 0.52±0.01 | 0.75±0.02 | 0.44±0.01 | 0.64±0.02 | 0.91±0.04 | 0.54±0.02 | 0.38±0.01 | 0.75±0.02 | 0.29±0.01 |
| | HITON + DD | 0.55±0.01 | 0.77±0.01 | 0.48±0.01 | 0.69±0.02 | 0.88±0.01 | 0.61±0.02 | 0.42±0.01 | 0.77±0.02 | 0.33±0.01 |
| 500 | HITON | 0.58±0.01 | 0.83±0.02 | 0.50±0.01 | 0.75±0.00 | 0.92±0.01 | 0.66±0.00 | 0.46±0.01 | 0.85±0.01 | 0.35±0.01 |
| | HITON + DD | 0.66±0.01 | 0.88±0.01 | 0.57±0.02 | 0.78±0.02 | 0.93±0.02 | 0.70±0.02 | 0.49±0.01 | 0.85±0.02 | 0.38±0.01 |
| 1000 | HITON | 0.71±0.01 | 0.90±0.01 | 0.64±0.01 | 0.78±0.02 | 0.99±0.00 | 0.67±0.02 | 0.48±0.01 | 0.87±0.01 | 0.36±0.01 |
| | HITON + DD | 0.72±0.01 | 0.88±0.01 | 0.65±0.01 | 0.83±0.02 | 0.99±0.00 | 0.74±0.03 | 0.52±0.01 | 0.88±0.01 | 0.40±0.01 |
| 2000 | HITON | 0.77±0.00 | 0.95±0.01 | 0.69±0.00 | 0.86±0.00 | 1.00±0.00 | 0.77±0.00 | 0.51±0.01 | 0.90±0.01 | 0.38±0.01 |
| | HITON + DD | 0.78±0.01 | 0.94±0.01 | 0.72±0.01 | 0.89±0.01 | 1.00±0.00 | 0.83±0.01 | 0.54±0.01 | 0.90±0.01 | 0.42±0.01 |

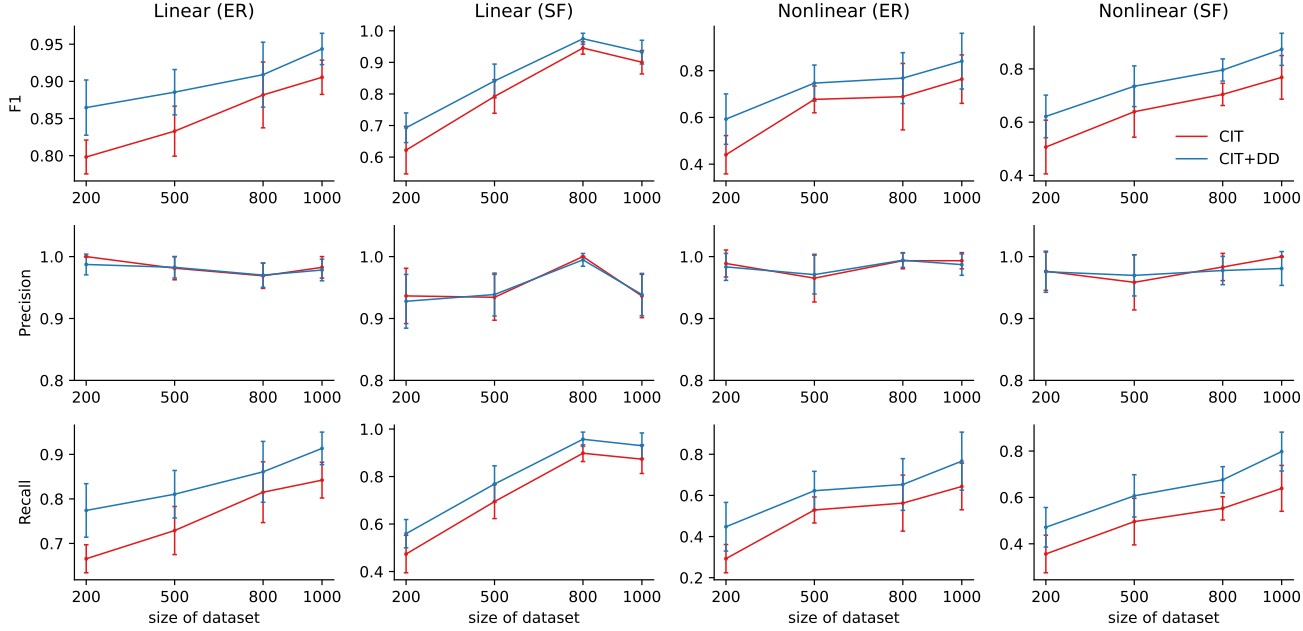

Figure 12: Results of correction experiments on continuous data with linear/non-linear relationships and different graph topologies, Erdös-Renyi (ER) and Scale-Free (SF). Error bars represent 95% confidence interval.

## C.3    RESULTS OF CORRECTION EXPERIMENT ON CONTINUOUS DATA

Although our experiments primarily focused on discrete data, it's worth noting that our method is versatile and applicable to various data types. To demonstrate this, we conducted additional evaluations using continuous data exhibiting both linear and non-linear relationships. We present the experimental results in Fig. 12. We constructed DAGs with 10 variables and 24 edges, adopting both Erdös-Renyi (ER) and Scale-Free (SF) models for DAG topologies. In parameterizing the Bayesian networks, we explored both linear and non-linear models. Specifically, we utilized multi-layer perceptrons to conduct non-linear parameterizations. For performing CIT, we employed partial correlation for linear settings and kernel-based CIT (KCI) [Zhang et al., 2012] for non-linear settings. We employed a Gaussian kernel for KCI and set kernel width using the median heuristic.[1]

Similar to our evaluations on discrete data, Fig. 12 illustrates that DEDUCE-DEP has the ability to effectively correct underpowered tests in a continuous data setting as well. This is evidenced by the increased recall and competitive performance in terms of precision. Notably, this improvement was consistently observed across different DAG topologies and Bayesian network parameterizations. This suggests that our method holds promise for a broader range of applications beyond discrete data analysis.

---

[1]We adopted KCI implementation from causal-learn Python package [Zheng et al., 2024].

# D RECURSIVE RELATIONSHIPS INDUCED BY DEDUCE-DEP

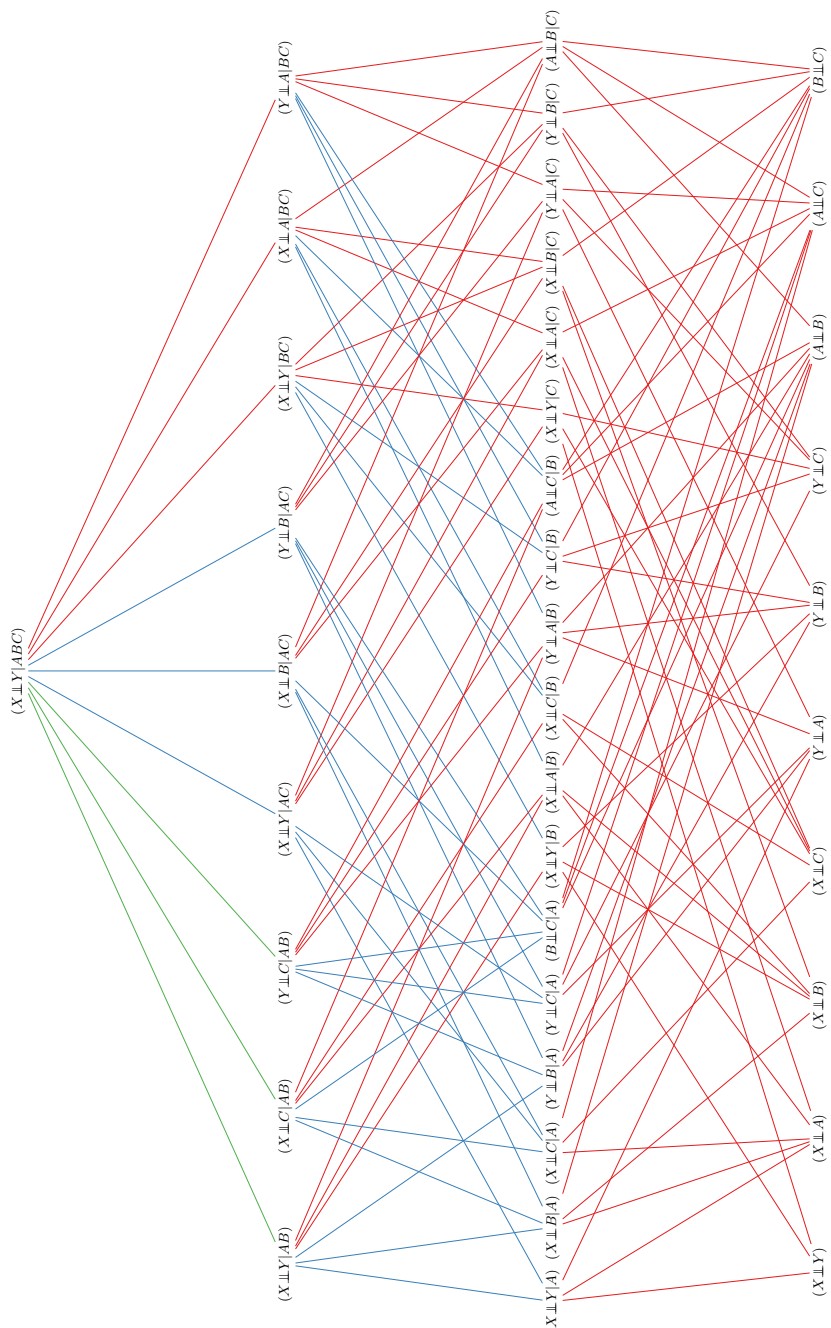

Figure 13: Recursive relationships induced by DEDUCE-DEP. Note, however, that not all recursion is necessary due to early stopping and/or conditional dependence. Given one higher-order test, the three one-less-order tests connected with the same colored edges form a single condition to deduce the dependence of the higher-order test.

In theory, the worst-case complexity of DEDUCE-DEP is exponential with respect to the worst-case depth. The worst-case scenario can be understood through a hypothetical call-tree exemplified in Fig. 13, where every CIT invokes recursive calls. However, we emphasize that this is generally not the case in practice. As shown in Fig. 6, the additional computational cost is not prohibitive due to cached previous CIT results and early stopping (i.e., not meeting conditions for deduction). As a side note, we can introduce a new hyperparameter $H$ which limits the maximum depth of recursion. The worst-case depth would then depend on $H$, involving the trade-off between efficiency (lower $H$) and accuracy (higher $H$).