# OpenReview forum: "Causal Discovery with Deductive Reasoning: One Less Problem"
_auai.org/UAI/2024/Conference — UAI 2024 poster_

### Official Review · Reviewer_Le4F · 2024-03-01

**Q2-1 Originality-Novelty:** 2
**Q2-2 Correctness-Technical Quality:** 3
**Q2-5 Clarity Of Writing:** 4

**Q1 Summary And Contributions:**

The paper describes a possible way of improving the structural accuracy of constraint-based learning algorithms for Bayesian networks. The key idea is to spot inconsistencies in the patterns of conditional independence tests used by the algorithms using the graphoid axioms. This approach is complementary to previous heuristic approaches proposed in the literature.

**Q2-3 Extent To Which Claims Are Supported By Evidence:**

2: Fair: the main claims are somewhat supported by evidence (but the experimental evaluation may be weak, or does not match entirely with the claims, important baselines may be missing, proofs contain important ideas but lack rigor, algorithmic details are only discussed superficially, references are imprecise, assumptions are not sufficiently motivated or explicated, etc.).

**Q2-4 Reproducibility:**

3: Good: key resources (e.g. proofs, code, data) are available and key details (e.g. proofs, experimental setup) are sufficiently well-described for competent researchers to confidently reproduce the main results.

**Q3 Main Strengths:**

The main strength of the paper is that using graphoid axioms provides a theoretically sound alternative to previous proposals, which are sensible but completely heuristic. Formal proofs are provided for all steps.

**Q4 Main Weakness:**

The main weakness of the paper is its experimental evaluation. The setup described in Section 5.1 does not cover a suitably wide range of scenarios (for instance, only DAGs with 20 nodes are considered), does not use any of the common evaluation metrics for Bayesian networks (SHD) or causal networks (SID), and has extremely simple parameterization (only binary variables). No evaluation of real-world data.

**Q5 Detailed Comments To The Authors:**

The theoretical and methodological construction of the paper is sound, with a clear objective and formal proofs for all derivations. The technical writing is also very clear. However, I find the experimental evaluation of the proposed methods quite weak. There are several ways in which it could be improved.

* The evaluation metrics (precision, recall and F1 score) could be substituted with SHD and SID which are specific to causal DAGs. This would allow for easier comparisons with the rest of the literature, and a more grounded evaluation of performance improvements.
* The authors do not provide sufficient details to understand how the topologies of the random DAGs are generated. More clarity on this matter would be good for reproducibility.
* The authors only consider random DAGs with 20 nodes and binary variables. Having multiple DAG sizes, different topologies, different arc densities, etc., as well as different parameterizations for the local distributions, is necessary to understand how the correction works in a general setting.
* I find it unlikely that evaluating random sets of conditional independence tests will be informative for real-world applications, the patterns of tests performed by structure learning algorithms will be very different. The authors should motivate better why the experiments are needed.

The authors themselves admit that "evaluating the effectiveness of our methodology solely based on this experiment is inadequate", but they spend 1 1/2 pages discussing it. I feel that space could have been put to better use.

The later evaluation using actual structure learning is more informative, but it is still limited by the choice of metrics. The computational complexity measure are also convincing and much appreciated.

The authors also say, "As the number of edges increases and the amount of data decreases, the difference in F1 scores ... becomes more pronounced." This trend should not be surprising: more arcs means more parameters, which means that conditional independence tests become more reliable for any fixed sample size. Type I and type II error rate will change as a function of the ratio #parameter/#samples, even more so with the typical asymptotic conditional independence test that I imagine the authors are using.

**Q9 Complying With Reviewing Instructions:**

Yes

---

> ### Author Rebuttal · Authors · 2024-04-07
>
> We sincerely appreciate your efforts and constructive comments to improve the manuscript. We respond to your comments below.
>
> ---
>
> > [Q1] The authors only consider random DAGs with 20 nodes / multiple DAG sizes / different arc densities / No evaluation of real-world data
> >
> - Due to the space constraints, experimental results on different DAG sizes ($|V|=10, 30$) and various arc densities ($|E|/|V| = 1.2,\\,1.5,\\, 2$) are provided in `Appendix B.2`.
> - Evaluations on real-world data (i.e., Alarm, Sachs, Insurance) are provided in `Fig 4`.
>
> > [Q2] topologies / only binary variables / different parameterizations
> >
> - In our manuscript, we used Erdös-Renyi random graph generation for the experiments on synthetic datasets. We will clarify this in the final version.
> - Following the reviewer’s suggestions, we provide additional experimental results on a different topology (i.e., scale-free random graph) and continuous variables. Please see our global response **[G1]**.
>
> > [Q3] The authors should motivate better why the [correction] experiments are needed.
> >
> - An intriguing property of our method is its *simple* and *modular* nature, which makes it possible to seamlessly integrate into any algorithm utilizing CITs. We demonstrated the effectiveness of our method on representative algorithms PC and HITON-PC, but it is infeasible to examine it on every existing algorithm.
> - The very motivation for the correction experiment is to **isolate the algorithm-dependent factors** and purely validate the intrinsic effectiveness of our method: *how well does our method correct the high-order CI statements?* For this, we use random sets of CI statements since the patterns of tests could be very different across different structure learning algorithms.
>
> > [Q4] The evaluation metrics (precision, recall and F1 score) could be substituted with SHD and SID which are specific to causal DAGs.
> >
> - Following the reviewer’s suggestion, we provide the additional evaluations measured as structural hamming distance (SHD); please see the global response **[G1]**.
> - The rationale behind the evaluation using F1 score is that our method intervenes in the skeleton learning step, and the rest of the steps remain the same for PC (and HITON-PC involves finding the neighbor set for a specific variable). Nevertheless, our method of correcting high-order CI statements should indeed lead to performance improvement in terms of SHD as well, and we agree with the reviewer that this would further strengthen our paper. Thanks for the valuable suggestions!
>
> > [Q5] The authors also say, "As the number of edges increases and the amount of data decreases, the difference in F1 scores ... becomes more pronounced.” [..] Type I and type II error rate will change as a function of the ratio #parameter/#samples
> >
> - As the reviewer pointed out, CITs indeed become more unreliable as the number of edges increases and the amount of data decreases; in fact, this is where our method comes into play with a greater advantage. The results in Table 1 validate the effectiveness of our method, and our analysis highlighted this outcome, which we had indeed expected.
>
> ---
>
> We will incorporate our responses and discussions into the revised manuscript. Thank you again for your detailed feedback.

---

### Official Review · Reviewer_QepW · 2024-03-16

**Q2-1 Originality-Novelty:** 2
**Q2-2 Correctness-Technical Quality:** 3
**Q2-5 Clarity Of Writing:** 3

**Q1 Summary And Contributions:**

The paper introduces a new method for performing more conservative constraint based causal discovery. Conditional independence tests have low power for large conditioning sets, and the authors show how lower order conditional independence statements can be used to deduce dependence among variables conditional on higher order sets. They show how this can be implemented in two different constraint based algorithms, and the theoretical results are supported by a simulation study.

**Q2-3 Extent To Which Claims Are Supported By Evidence:**

3: Good: the main claims are supported by convincing evidence (in the form of adequate experimental evaluation, proofs, (pseudo-)code, references, assumptions).

**Q2-4 Reproducibility:**

3: Good: key resources (e.g. proofs, code, data) are available and key details (e.g. proofs, experimental setup) are sufficiently well-described for competent researchers to confidently reproduce the main results.

**Q3 Main Strengths:**

The paper aims to resolve an existing problem with a simple solution, and it appears to be technically sound. In the simulation study, they find that the method in general improves the recall with only a small decrease in the precision (reflected in an increased F1 score).

**Q4 Main Weakness:**

(1) The content is limited: The method is quite simple and the proofs seem somewhat straightforward. The paper doesn't introduce any other theoretical insights, only a few examples, and limited discussion.

(2) The writing and organization of the paper need some improvement:

- Some of the text seems superfluous, and there is a lot of unnecessary repetition:  E.g., they keep mentioning the fact that they only use low order tests.

- The authors sometimes write "CIT" and sometimes "CI tests", which can be confusing.

- The term "reliability criterion" is defined in Section 3, but already used in Section 2.2

**Q5 Detailed Comments To The Authors:**

(1) In light of Q4, I would suggest the authors to omit some of the repeated sentences and instead include, e.g., another example, figure etc., bringing more new content to the paper.

(2)  I'm curious how this method works combined with the criteria mentioned in Section 2.2.

(3) The authors don't discuss how to find the optimal value of the minimal conditioning set size K. It would have been interesting if the authors had discussed this, or if they had used different values of K in the simulation study.

(3) Typos:

- In Section 5.2 the authors refer to Table 4, but I think they mean Table 1.

**Q9 Complying With Reviewing Instructions:**

Yes

---

> ### Author Rebuttal · Authors · 2024-04-07
>
> Thank you for your insightful comments. We greatly appreciate your valuable input and address your comments as follows.
>
> ---
>
> > [Q1] The content is limited: The method is quite simple and the proofs seem somewhat straightforward.
> >
> - The PC algorithm has been around for more than 30 years now, and it has been at least about 20 years since heuristics were considered to address the reliability issues of CITs. Yet, existing works have relied on complex heuristics, and to the best of our knowledge, none of them have been as simple, algorithm-agnostic, and easily applicable as ours. We believe the *simplicity* and *modular* nature of our method is a key strength of our work: It is easy to use and can be seamlessly integrated with any algorithm utilizing CITs.
>
> > [Q2] I'm curious how this method works combined with the criteria mentioned in Section 2.2.
> >
> - It could lead to performance improvement, but this may not always be the case since the reliability criterion distorts the structure learning process by omitting CITs that are necessary to yield causal structure. Nevertheless, we agree that this analysis would be interesting and we will include it in the final manuscript.
>
> > [Q3] the minimal conditioning set size K / different values of K
> >
> - We provide the experimental results across various values of K (see global response **[G1]**): the higher value of K leads to computational efficiency, whereas the lower value of K leads to performance improvement. Determining such trade-offs would depend on the practitioners’ interests and domain knowledge. We believe it is also an intriguing property of our method to offer such flexible controllability.
>
> > [Q4] The writing and organization of the paper / typos
> >
> - We deeply appreciate the reviewer’s constructive feedback. We will carefully revise the paper to improve the clarity based on the comments from the reviewer.
>
> ---
>
> We will incorporate our responses and discussions into the revised manuscript. Thank you again for your detailed feedback.

---

### Official Review · Reviewer_Uu7f · 2024-03-22

**Q2-1 Originality-Novelty:** 2
**Q2-2 Correctness-Technical Quality:** 3
**Q2-5 Clarity Of Writing:** 4

**Q1 Summary And Contributions:**

The paper presents a simple and straight-forward correction method for unreliable CITs in constraint-based causal discovery based on rules from graphoid axioms.

**Q2-3 Extent To Which Claims Are Supported By Evidence:**

4: Excellent: all claims are supported by very convincing evidence (in the form of comprehensive experimental evaluation, rigorous mathematical proofs, detailed (pseudo-)code, precise references, well-motivated and realistic assumptions) and the authors deliver what they promise.

**Q2-4 Reproducibility:**

3: Good: key resources (e.g. proofs, code, data) are available and key details (e.g. proofs, experimental setup) are sufficiently well-described for competent researchers to confidently reproduce the main results.

**Q3 Main Strengths:**

The simple and modular nature of the correction method is a clear strength. Falsely removed edges in constraint-based structure learning propagate through the algorithm and detecting and correcting unreliable CITs seems to improve performance quite substantially.

**Q4 Main Weakness:**

The paper seems to be written with only discrete data in mind (see reliability heuristics). What heuristics exist in the continuous setting and in case of the PC-Algorithm, lower order CITs have already been conducted so that it would make sense to always use the correction I assume (I think you say this somewhere as well)? Is there something to say then about worst case computational complexity?

You don't refer to any open-source implementation. I assume it wouldn't be hard to modulate this into existing software, but I would prefer if you provide this.

**Q5 Detailed Comments To The Authors:**

Just some minor things:

Figure 3's caption is a bit short and it would be good to add explanations there directly.

In Section 5.2. your hyperlink to table 1 points to table 4 in the appendix.

**Q9 Complying With Reviewing Instructions:**

Yes

---

> ### Author Rebuttal · Authors · 2024-04-07
>
> We sincerely appreciate the reviewer’s time and efforts to provide constructive feedback to improve our paper. We respond to each of your comments below.
>
> ---
>
> > [Q1] The paper seems to be written with only discrete data in mind (see reliability heuristics). What heuristics exist in the continuous setting
> >
> - Reliability heuristics have been prevalent for discrete data, e.g., since the test statistics often involve counts of data points. Although we mainly focused on discrete settings, our method is indeed agnostic of the data type. We provide additional evaluations on continuous data with linear/non-linear relationships. Please see the global response **[G1]**.
>
> > [Q2] in case of the PC-Algorithm, lower order CITs have already been conducted so that it would make sense to always use the correction I assume? / worst case computational complexity
> >
> - The PC algorithm starts by performing low-order CITs and eventually performs high-order CITs. Our method delegates such high-order CI statements to (more reliable) low-order CITs, while efficiently utilizing cached previous CIT results.
>
> - In theory, the worst-case complexity is exponential with respect to the worst-case depth. The worst-case scenario can be understood through a hypothetical call-tree exemplified in Fig. 9, where every CIT invokes recursive calls. However, we emphasize that this is generally not the case in practice. As shown in Fig. 8, the additional computational cost is not prohibitive due to cached previous CIT results and early stopping (i.e., not meeting conditions for deduction).
> - As a side note, we can introduce a new hyperparameter (say, H) which limits the maximum depth of recursion. The worst-case depth would then depend on H, involving the trade-off between efficiency (lower H) and accuracy (higher H).
>
> > [Q3] open-source implementation
> >
>
> - Please see the global response **[G2]**.
>
> > [Q4] Figure 3's caption / hyperlink to table 1
> >
> - Sorry for the inconvenience. We will fix the hyperlink and add explanations in the caption in the final version.
>
> ---
>
> We will incorporate our responses and discussions into the revised manuscript. Thank you again for your detailed feedback.

---

### Official Review · Reviewer_dxdr · 2024-03-22

**Q2-1 Originality-Novelty:** 3
**Q2-2 Correctness-Technical Quality:** 3
**Q2-5 Clarity Of Writing:** 3

**Q1 Summary And Contributions:**

This paper proposes DEDUCE-DEP to mitigate the possible error brought by a large conditional set by employing deductive reasoning based on conditional independence (CI) statements and graphical axioms to replace higher-order CIT test results. The performance shows promise in high recall with a mild drop in precision but overall achieves F1.

**Q2-3 Extent To Which Claims Are Supported By Evidence:**

3: Good: the main claims are supported by convincing evidence (in the form of adequate experimental evaluation, proofs, (pseudo-)code, references, assumptions).

**Q2-4 Reproducibility:**

3: Good: key resources (e.g. proofs, code, data) are available and key details (e.g. proofs, experimental setup) are sufficiently well-described for competent researchers to confidently reproduce the main results.

**Q3 Main Strengths:**

Strengths:
1. This paper introduces a deductive rule based on graphoid axioms allowing to replace the higher-order CI statements with lower-order CI test (smaller Z sets).
2. The proposed method is easier to use, with a clearer logic, compared to older methods.
3. The paper is clear and well-structured.

**Q4 Main Weakness:**

Weaknesses:
1. It is unclear whether Proposition 1 is complete. How can this be demonstrated?
2. The basic idea of this work is to trust the CI test with a lower order. Can you provide some intuition on how the proposed method performs better than a PC method that simply tests and trusts the CI relation start from the low order conditional set.
3. Some experiment details are missing. What is the as-is.
4. It seems that the experiment only performs on discrete data. How is the performance on linear and nonlinear data?

**Q5 Detailed Comments To The Authors:**

See the weaknesses above.

**Q9 Complying With Reviewing Instructions:**

Yes

---

> ### Author Rebuttal · Authors · 2024-04-07
>
> We sincerely appreciate for reviewing our paper and providing us with constructive feedback. We respond to each of your comments below:
>
> ---
>
> > [Q1] Can you provide some intuition on how the proposed method performs better than a PC method that simply tests and trusts the CI relation start from the low order conditional set.
> >
> - We first note that PC eventually performs high-order CITs which suffer from low statistical power as the structure learning proceeds. In contrast, our method delegates such high-order CI statements to (more reliable) low-order CITs. As a side note, if the PC algorithm performs *only* low-order CITs, the structure would not be fully specified.
>
> > [Q2] How is the performance on linear and nonlinear data?
> >
> - Although we mainly focused on discrete settings, our method is indeed agnostic of the data type. Following the reviewer’s suggestions, we provide additional evaluations on continuous data with linear/non-linear relationships. Please see the global response **[G1]**.
>
> > [Q3] It is unclear whether Prop 1 is complete. How can this be demonstrated?
> >
> - We would like to clarify that our method utilizes Corollary 1 (Line 10 in Algorithm 1), where its completeness for deducing the dependence statement is established in Prop 2.
>
> > [Q4] Some experiment details are missing. What is the as-is.
> >
> - Sorry for the confusion. “as-is” denotes the vanilla algorithms *without* deduce-dep. We will clarify this in the final version.
>
> ---
>
> We will incorporate our responses and discussions into the revised manuscript. Thank you again for your detailed feedback.

---

### Official Review · Reviewer_2K6f · 2024-03-22

**Q2-1 Originality-Novelty:** 3
**Q2-2 Correctness-Technical Quality:** 3
**Q2-5 Clarity Of Writing:** 3

**Q1 Summary And Contributions:**

In the context of constraint-based causal discovery, Conditional Independence Tests (CIT) are of paramount importance.  The paper proposes a principled, simple, yet effective method, which corrects unreliable independence statements by replacing them with deductively reasoned results from lower-order CITs. The paper also provide as empirical evaluation.

I acknowledge to have read the authors rebuttal.

**Q2-3 Extent To Which Claims Are Supported By Evidence:**

3: Good: the main claims are supported by convincing evidence (in the form of adequate experimental evaluation, proofs, (pseudo-)code, references, assumptions).

**Q2-4 Reproducibility:**

2: Fair: key resources (e.g. proofs, code, data) are unavailable but key details (e.g. proof sketches, experimental setup) are sufficiently well-described for an expert to confidently reproduce the main results.

**Q3 Main Strengths:**

- a simple method to correct unreliable independence statements by replacing them with deductively reasoned results from lower-order CITs

**Q4 Main Weakness:**

- neither code nor data has been provided to validate the experimental claims. To me, it's like a main theorem without (sketched) proof. So numbers in Table 1 are not verifiable

- why F1 measure only? What about also accuracy or other effectiveness measures?

- I still miss why recall increased so much, while precision decreases. This phenomena needs a better explanation

- what is the computational complexity of Deduce-Dep ?

**Q5 Detailed Comments To The Authors:**

Paper is reasonably well written. I've some concern related to Q4 that need to be addressed.

minor:
- in the motivating example, anticipate the meaning of the tuples

**Q9 Complying With Reviewing Instructions:**

Yes

---

> ### Author Rebuttal · Authors · 2024-04-07
>
> We appreciate the reviewer’s valuable efforts and instructive advice to improve the manuscript. We respond to your comments as follows:
>
> ---
>
> > [Q1] I still miss why recall increased so much, while precision decreases. This phenomena needs a better explanation.
> >
> - Thanks for the question. This phenomenon aligns with the key motivation of our work: Through deductive reasoning, we retain true edges at risk of being removed by CITs with low statistical power. Specifically, we can expect an improvement in recall in retaining true edges, while precision may decrease if there exist errors in low-order CITs, which is less likely. A significant increase in F1 score (with a huge increase in recall and minor compensation in precision) demonstrates the effectiveness of our method that retains more true edges than false ones.
>
> > [Q2] why F1 measure only? What about also accuracy or other effectiveness measures?
> >
> - The rationale behind our evaluation with the F1 score (+ precision, recall) relates to the key motivation of our work described above. Following the reviewer’s suggestion, we provide the additional evaluations measured as structural hamming distance (SHD); please see the global response **[G1]**.
>
> > [Q3] what is the computational complexity of Deduce-Dep?
> >
>
> - In theory, the worst-case complexity is exponential with respect to the worst-case depth. The worst-case scenario can be understood through a hypothetical call-tree exemplified in Fig. 9, where every CIT invokes recursive calls. However, we emphasize that this is generally not the case in practice. As shown in Fig. 8, the additional computational cost is not prohibitive due to cached previous CIT results and early stopping (i.e., not meeting conditions for deduction).
> - As a side note, we can introduce a new hyperparameter (say, H) which limits the maximum depth of recursion. The worst-case depth would then depend on H, involving the trade-off between efficiency (lower H) and accuracy (higher H).
>
> > [Q4] neither code nor data has been provided to validate the experimental claims.
> >
>
> - Please see the global response **[G2]**.
>
> ---
>
> We will incorporate our responses and discussions into the revised manuscript. Thank you again for your detailed feedback.

---

### Meta-Review · Area_Chair_A8up · 2024-04-16

The paper is a solid contribution with no major flaws. While it only provides a moderate amount of theoretical novelty, it has significant practical importance. The authors propose a module that can replace statistical tests of conditional independence that have a large conditioning set with logically implied statistical tests of conditional independence that have a smaller conditioning set, thereby increasing the power of the statistical tests. This module can be inserted into any structural learning algorithm that relies on conditional independence tests, which covers a lot of different algorithms. They demonstrate that their module greatly increases recall and F1 on both randomly chosen conditional independence tests and on the structure of DAGs output by causal structure learning algorithms, with only a slight decrease in precision. The improvements are quite substantial and should be useful in practice.

There was generally a consensus among the reviewers about the quality of the paper and all of the reviewers were confident in their views. All of the reviewers recommended publication, ranging from “borderline accept” to “accept”, and averaging “weak accept”. One of the weaker points is related to novelty. 3 of the reviewers considered novelty only “fair”, with the others rating it as “good”. I agree that that the novelty is not high, but feel that is made up for by the likely practical usefulness of the contribution. Several weaknesses of the paper identified in the reviews have been addressed in the discussion. While the simulations demonstrating the improvements that occurred when using the proposed module were originally narrow in focus (only binary variables) they have since expanded the simulations to include linear and non-linear cases, and shown that the additional computational costs are not burdensome. They have also provided access to their code.